# Yersiniomics, a Multi-Omics Interactive Database for *Yersinia* Species

Pierre Lê-Bury,[a] Karen Druart,[b] Cyril Savin,[a,e] Pierre Lechat,[c] Guillem Mas Fiol,[a] Mariette Matondo,[b] Christophe Bécavin,[d] Olivier Dussurget,[a] Javier Pizarro-Cerdá[a,e]

aInstitut Pasteur, Université Paris Cité, CNRS UMR6047, *Yersinia* Research Unit, Paris, France
bInstitut Pasteur, Université Paris Cité, CNRS USR2000, Mass Spectrometry for Biology Unit, Proteomic Platform, Paris, France
cInstitut Pasteur, Université Paris Cité, ALPS, Bioinformatic Hub, Paris, France
dUniversité Côte d'Azur, CNRS, IPMC, Sophia-Antipolis, France
eInstitut Pasteur, Université Paris Cité, *Yersinia* National Reference Laboratory, WHO Collaborating Research & Reference Centre for Plague FRA-140, Paris, France

**ABSTRACT** The genus *Yersinia* includes a large variety of nonpathogenic and life-threatening pathogenic bacteria, which cause a broad spectrum of diseases in humans and animals, such as plague, enteritis, Far East scarlet-like fever (FESLF), and enteric redmouth disease. Like most clinically relevant microorganisms, *Yersinia* spp. are currently subjected to intense multi-omics investigations whose numbers have increased extensively in recent years, generating massive amounts of data useful for diagnostic and therapeutic developments. The lack of a simple and centralized way to exploit these data led us to design Yersiniomics, a web-based platform allowing straightforward analysis of *Yersinia* omics data. Yersiniomics contains a curated multi-omics database at its core, gathering 200 genomic, 317 transcriptomic, and 62 proteomic data sets for *Yersinia* species. It integrates genomic, transcriptomic, and proteomic browsers, a genome viewer, and a heatmap viewer to navigate within genomes and experimental conditions. For streamlined access to structural and functional properties, it directly links each gene to GenBank, the Kyoto Encyclopedia of Genes and Genomes (KEGG), UniProt, InterPro, IntAct, and the Search Tool for the Retrieval of Interacting Genes/Proteins (STRING) and each experiment to Gene Expression Omnibus (GEO), the European Nucleotide Archive (ENA), or the Proteomics Identifications Database (PRIDE). Yersiniomics provides a powerful tool for microbiologists to assist with investigations ranging from specific gene studies to systems biology studies.

**IMPORTANCE** The expanding genus *Yersinia* is composed of multiple nonpathogenic species and a few pathogenic species, including the deadly etiologic agent of plague, *Yersinia pestis*. In 2 decades, the number of genomic, transcriptomic, and proteomic studies on *Yersinia* grew massively, delivering a wealth of data. We developed Yersiniomics, an interactive web-based platform, to centralize and analyze omics data sets on *Yersinia* species. The platform allows user-friendly navigation between genomic data, expression data, and experimental conditions. Yersiniomics will be a valuable tool to microbiologists.

**KEYWORDS** *Yersinia*, genome, transcriptome, proteome, database, RNA-Seq, mass spectrometry, microarray, multi-omics, synteny

The genus *Yersinia* comprises 26 Gram-negative bacterial species which belonged to the family *Enterobacteriaceae* until 2016 and which are now part of the new family *Yersiniaceae* (1). Although this genus includes mostly nonpathogenic environmental species, several important animal and human pathogens are also present in the group. *Yersinia enterocolitica* and *Yersinia pseudotuberculosis* are phylogenetically distant *Yersinia* species (2); however, the parallel acquisition of a diverse set of virulence factors (invasins, siderophores, and a type III secretion system) has endowed both species with the capacity

Address correspondence to Javier Pizarro-Cerdá, javier.pizarro-cerda@pasteur.fr, or Olivier Dussurget, olivier.dussurget@pasteur.fr.

The authors declare no conflict of interest.

to invade the gastrointestinal tract of mammals and to cause enteritis, following an oro-fecal infectious cycle (3). Enteric yersiniosis is the third most reported bacterial foodborne zoonosis in Europe (4). In the United States, *Y. enterocolitica* is classified as a priority pathogen by the National Institutes of Health (NIH). In Africa, recent evidence indicates that *Y. enterocolitica* causes human digestive disorders with a frequency similar to that reported in other continents (5). *Yersinia pestis*, on the other hand, is a clone that recently emerged from *Y. pseudotuberculosis* (6) and acquired the capacity to infect fleas and to cause plague in humans through acquisition of novel virulence factors and massive gene inactivation (7). Plague is still endemic in the Americas, Africa, and Asia (8), and the major pneumonic plague outbreak in Madagascar in 2017 is a reminder that *Y. pestis* is a severe threat to human populations (9, 10). Two other animal pathogens, *Yersinia ruckeri* and *Yersinia entomophaga*, have the capacity to cause disease in fishes and insects, respec-tively (11, 12). *Y. ruckeri* is responsible for enteric redmouth disease, one of the most im-portant disease of salmonids, which leads to significant economic losses (11). *Y. entomo-phaga* has a commercial interest for pest management, as it can infect and kill a wide range of insects (13).

Pathogenic *Yersinia* spp. have been instrumental models to understand the evolution and mechanisms of pathogenicity in the bacterial world. The invasin of *Y. pseudotubercu-losis* was the first bacterial factor reported to promote bacterial internalization within mammalian nonphagocytic epithelial cells (14, 15). The subsequent identification of $\beta$1 integrins as receptors for invasion (16) set the general basis to understand how bacterial surface effectors subvert mammalian cellular functions (phosphoinositide metabolism, Rho GTPase signaling, and actin polymerization) to invade host cells and tissues (17–20). The *Yersinia* type III secretion system was one of the first to be thoroughly characterized, making it possible to decipher the exquisite manipulation of phagocytic and immune functions by a bacterial pathogen through injection of bacterial effectors within the cyto-plasm of host neutrophils, macrophages, and dendritic cells (21, 22).

Like several other important bacterial pathogens, members of the genus *Yersinia* have been investigated using omics approaches. In the last 2 decades, the ever-grow-ing pace of technological innovation led to generation of a massive amount of data produced by omics methods, such as whole-genome sequencing combining short (such as Illumina sequencing) and long (such as PacBio or Nanopore sequencing) reads, DNA hybridization array (macro- and microarrays) for gene expression analysis, RNA sequenc-ing (RNA-Seq), and semiquantitative mass spectrometry (Fourier-transform ion cyclotron resonance [FTICR-MS] and liquid chromatography-tandem mass spectrometry [LC-MS/MS]). Data type-specific databases allowing the deposition of these data have been cre-ated, e.g., GenBank (23) for genomes, Gene Expression Omnibus (24) and ArrayExpress (25) for microarrays, European Nucleotide Archive (26) and Sequence Read Archive (27) for RNA-Seq, and ProteomeXchange (28) for mass spectrometry. However, integration of these data in a genus- or species-dependent manner is currently restricted to a few model organisms (29–31), and pathogenic-microorganism-specific integrated data found in databases like PATRIC (32) have not been updated with recent experiments.

Here, we present a unique curated multi-omics database gathering 200 genomic, 317 transcriptomic, and 62 proteomic data sets originating from *Yersinia* spp. since the beginning of the omics revolution. This database was constructed using the Bacnet platform (33); we contributed to improvements of this platform. Indeed, with omics technologies evolving rapidly, new data formats, such as LC-MS/MS shotgun, which was absent from the first Bacnet-based website, were implemented (31). For several reference genomes, we implemented integrated views at the gene level of the Kyoto Encyclopedia of Genes and Genomes (KEGG) for biochemical pathways (34), UniProt for protein information (35), InterPro for protein domain signature (36), IntAct for mo-lecular interactions (37) and the Search Tool for the Retrieval of Interacting Genes/Proteins (STRING) for protein interactions (38). Raw reads from 425 RNA-Seq runs were consistently processed and analyzed with our bioinformatic pipeline, and a quality con-trol and differential analysis report is available for each experiment which

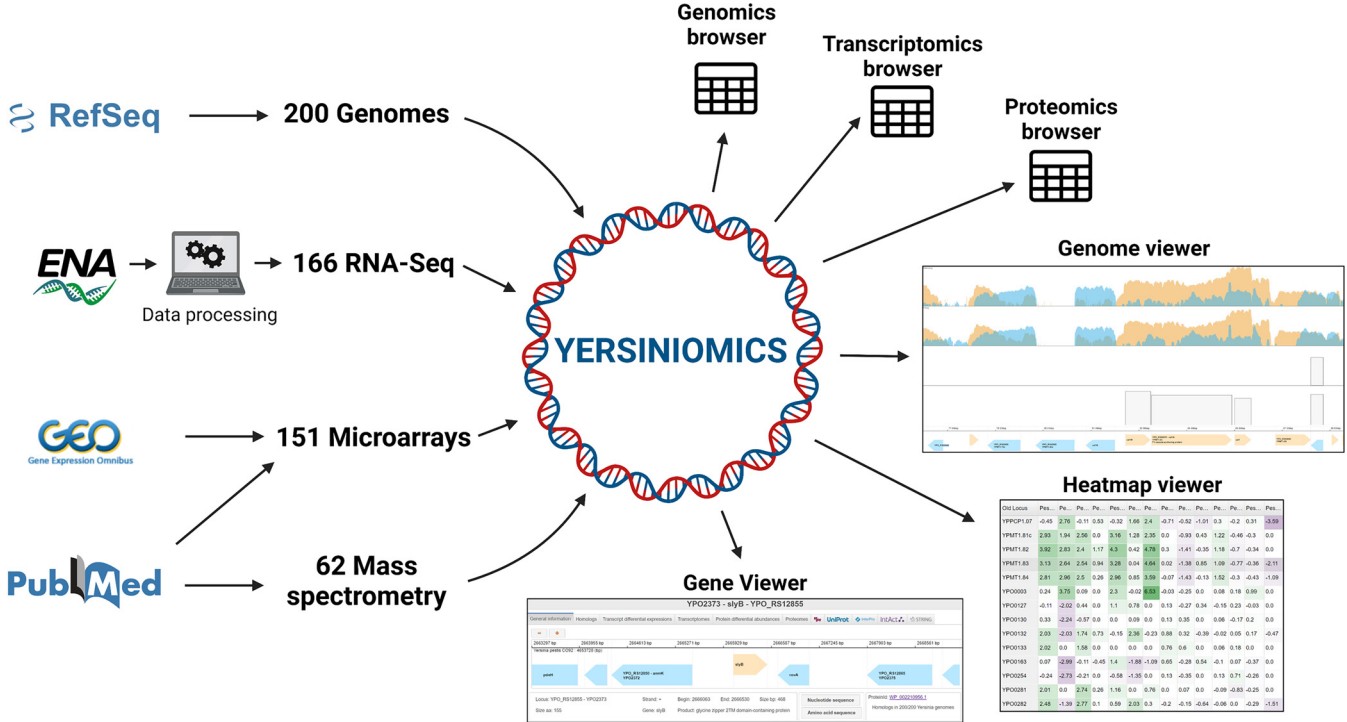

**FIG 1** Yersiniomics database construction pipeline and tools. Genomes were collected on the RefSeq database (GenBank). RNA-Seq raw data (.fastq files) were collected on the ENA browser and then processed as read counts and fold change. Fold changes from microarray experiments were collected on GEO or from tables and supplemental tables in publications. Fold changes from proteomic experiments were collected from tables and supplemental tables in publications. Data sets were integrated into Yersiniomics and can be consulted via three omics browsers (genomics, transcriptomics, and proteomics) and three viewers (gene, genome, and heatmap).

encompassed replicates. Processed omics experiments are easily browsable, linked to their Gene Expression Omnibus (GEO) (24), European Nucleotide Archive (ENA) (26) or Proteomics Identifications Database (PRIDE) (39) repositories, and directly viewable at the gene level or according to experimental conditions on the dedicated Yersiniomics website that we have implemented (https://yersiniomics.pasteur.fr/).

## RESULTS

From public databases, we collected genomic sequences, raw RNA-Seq data, microarray data, and mass spectrometry data that we processed and integrated into the Yersiniomics website (https://yersiniomics.pasteur.fr/). Yersiniomics relies on 6 principal tools (Fig. 1): (i) three omics browsers ("genomics," "transcriptomics," and "proteomics") in a table format allowing navigation among the different genomes and biological conditions of the experiments implemented in the database; (ii) two omics data set viewers, called "genome viewer" and "heatmap viewer," allowing navigation among transcriptomics and proteomics results; and (iii) a gene viewer allowing navigation among the genes of a specific genome, to quickly access associated entries in external databases and to browse associated omics data.

**Database functionalities. (i) The omics data set browsers.** From the Yersiniomics home page, three omics browsers are available: genomics, transcriptomics, and proteomics (Fig. 2, middle panel).

From the genomics browser, 200 complete *Yersinia* genomes are sorted according to their phylogenetic relatedness, based on the 500 genes of our recently proposed *Yersinia* core genome multilocus sequence typing (cgMLST) scheme (40) (Fig. 3). Displayed information related to genomes includes species name, the most recent assignation proposed by the French *Yersinia* National Reference Center and determined via our cgMLST, the number of chromosomes and plasmids, the chromosome and plasmid total size, the lineage and sublineage (when applicable) based on single nucleotide polymorphisms (SNP) analysis for

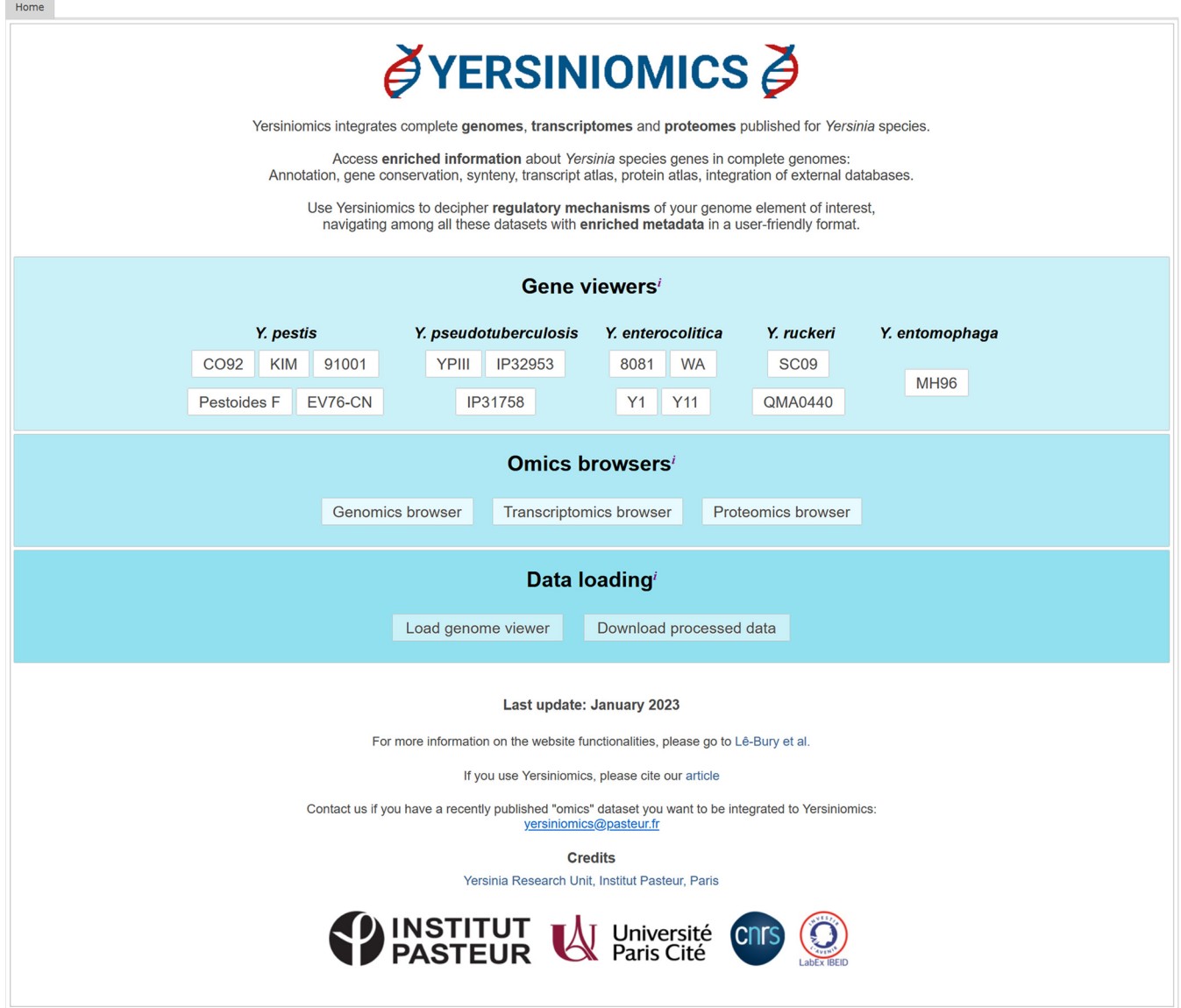

**FIG 2** Yersiniomics home page. In the top panel, shortcuts to strain gene viewers of *Y. pestis*, *Y. pseudotuberculosis*, *Y. enterocolitica*, *Y. ruckeri*, and *Y. entomophaga* are implemented. In the middle panel, three buttons lead to the genomics, transcriptomics, and proteomics browsers. The bottom panel allows the user to load previously saved genome viewers and to download omics processed data.

*Y. pestis* (41, 42) and on our cgMLST (40) for the other *Yersinia* species, the biotype and serotype (when relevant), the isolation source, year and country, the number of genes, rRNAs, and tRNAs, and the assembly ID, linked to GenBank, and its FTP link. A specific genome can be opened in the gene viewer (see below) by double clicking on it, and several genomes can be selected to export a summary table. Genomes are highlighted in blue in the table when searched with the search box, and genomes selected in the table are highlighted in red on the phylogenetic tree displayed on the left, which can be exported in SVG format (Fig. 3).

The transcriptomics and proteomics browsers work in a similar way and summarize the biological conditions of transcriptomic and proteomic *Yersinia* experiments (Fig. 4). Biological condition names follow a nomenclature encompassing the main important information about each experiment: species name, strain name, whether it addresses a wild-type (WT) or a mutant strain (when applicable), culture temperature, culture medium/phase/time point (when applicable), other information relevant to comparisons (when applicable), the technology used, and the year of data deposition or publication. For example, "Pestis_KIM6+_WT_37C_HIB_pH6_NextSeq500_2021" refers to an

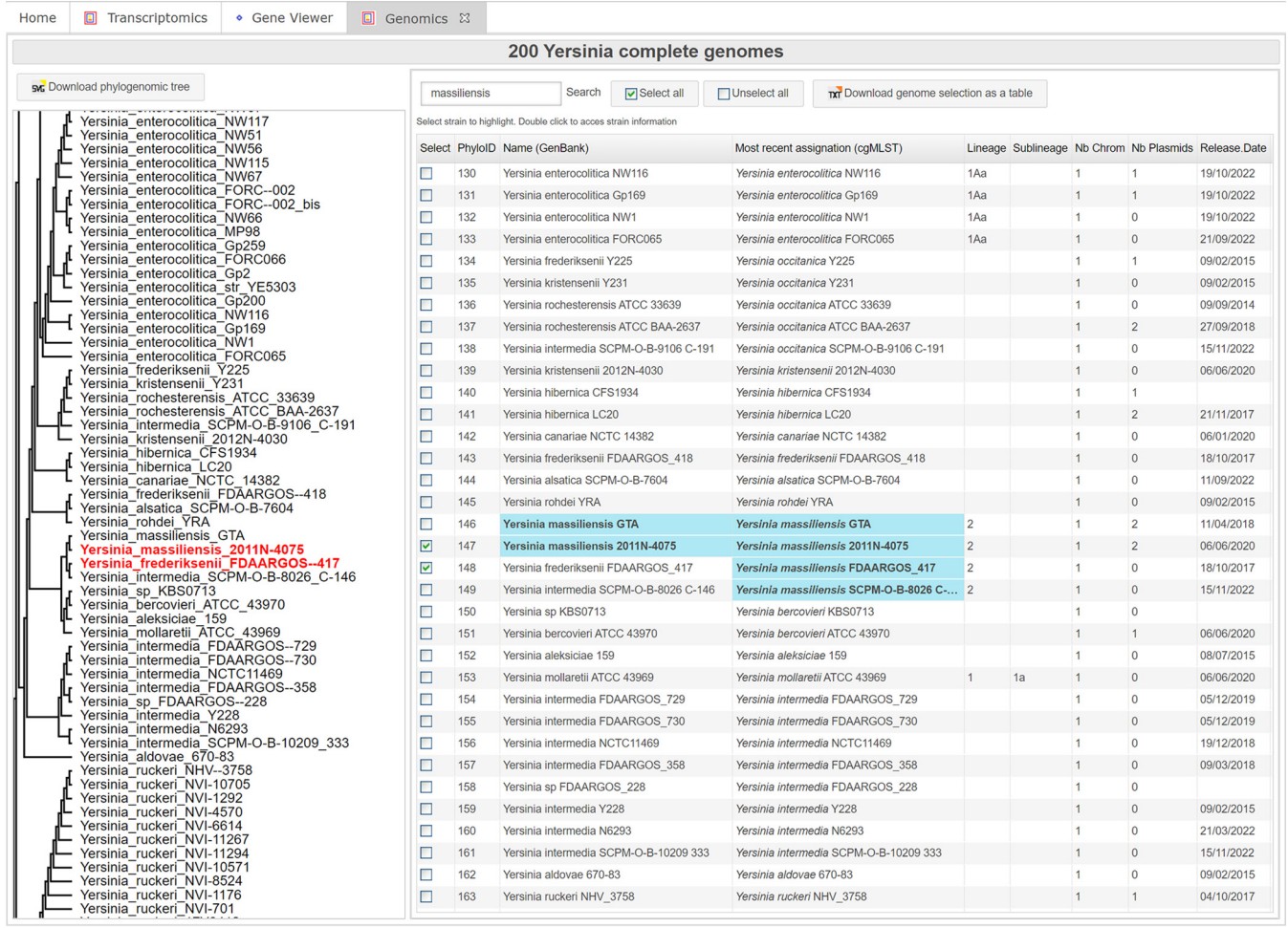

**FIG 3** The genomics browser. A *Yersinia* phylogenetic tree is displayed on the left panel, and the strains information (name, lineage, isolate source, number of genes, etc.) is available on the right panel as a searchable table. Selected strains are automatically highlighted on the phylogenetic tree. An updated tree can be downloaded in SVG format.

RNA-Seq experiment using the Illumina NextSeq500 platform deposited in 2021 and measuring RNA levels of a *Y. pestis* KIM6+ strain grown at 37°C in heart infusion broth (HIB) at pH 6. "Pseudotuberculosis_YPIII_Mutant_rovA_25C_Log_Agilent_2014" refers to an experiment performed with an Agilent microarray in 2014 using a *Y. pseudotuberculosis* YPIII strain mutated at the *rovA* locus and grown at 25°C to logarithmic phase. Complementary information, such as details on the strain microarray, the genome on which the RNA-Seq experiment was mapped, the DESeq2 (43) differential analysis report for RNA-Seq data, the publication reference with a link to PubMed, and the link to the GEO (24), ENA (26), or PRIDE (39) entry, can be found in the table columns. A search bar can be used to explore the biological conditions of each experiment, and some filters are accessible, such as the reference genome used, the data type ("Gene Expression" for microarray experiments or "RNA-Seq"), as well as the selection of a specific mutant or growth phase. The proteomics browser also implements a "protein localization" column (i.e., supernatant, whole cell, or soluble or insoluble fraction), as well as the number of proteins detected in each experimental condition.

After selecting one or several biological conditions, their associated transcriptome or proteome values can be visualized in two ways: as a heatmap viewer or as a genome viewer. The heatmap viewer is a sortable table with each line corresponding to a gene in the genome and columns displaying expression comparisons of two biological conditions using a color code proportional to the fold change, facilitating tracking of highly differentially expressed genes. A $\log_2$(fold change) cutoff, set to 1.5 by default,

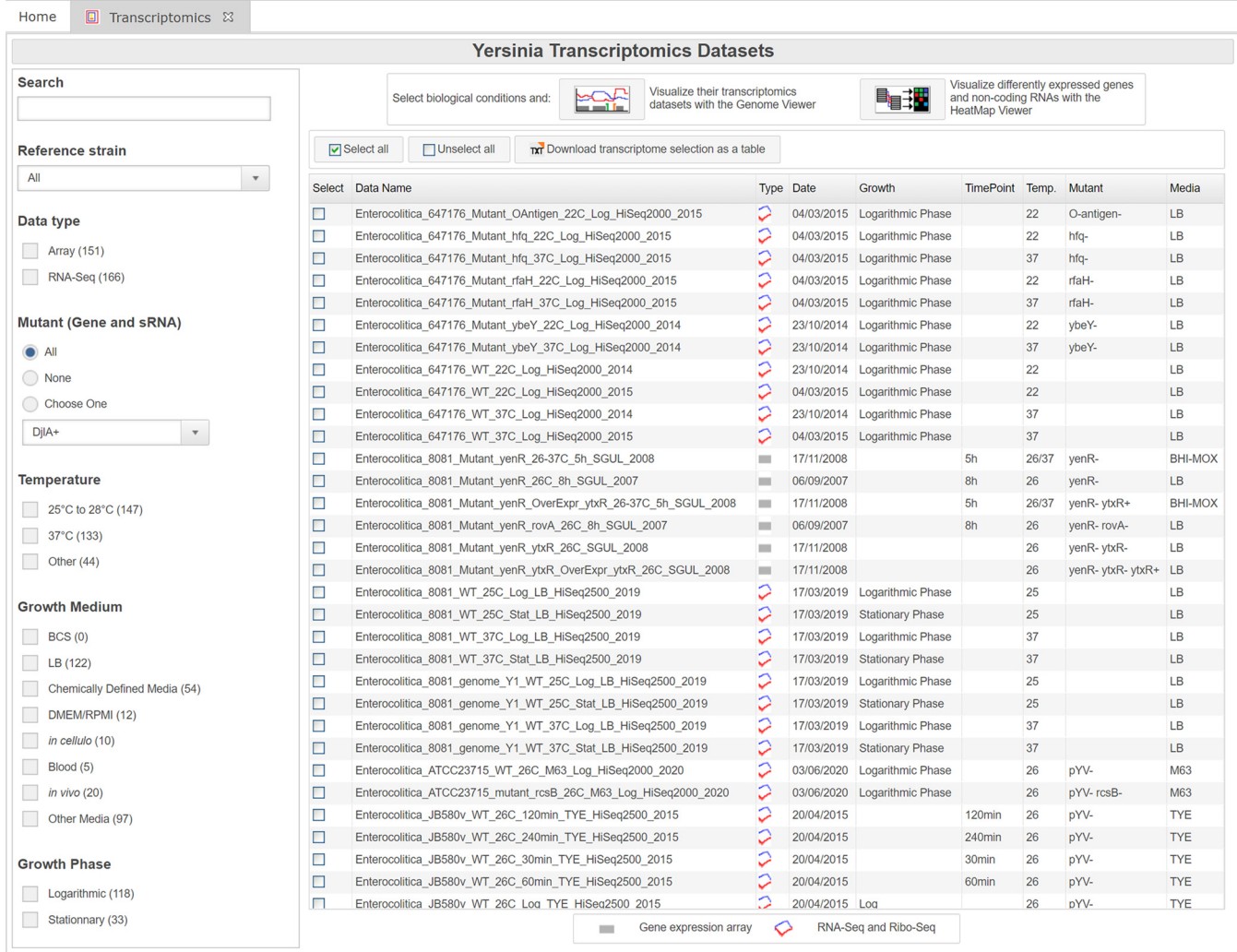

**FIG 4** The transcriptomics browser. Transcriptomics data sets are browsable in the main panel on the right and can be filtered through the search bar or with the preset filters on the left. Information on biological conditions, such as culture temperature, medium, growth stage, strain used, mutation, associated publication, and DESeq2 differential analysis report for RNA-Seq experiments, is accessible in the different columns of the table. After one or several lines are selected, the associated biological conditions can be viewed in the genome viewer or in the heatmap viewer by clicking on the buttons at the top.

can be set to any other value, or to 0 to display the whole genome. This view allows comparison of different results such as *Y. pestis* RNA abundance in infected C57BL/6 mouse lungs versus growth in brain heart infusion broth (BHI) (44), *Y. pestis* RNA abundance after growth in human plasma versus growth in lysogeny broth (LB) (45), and *Y. pestis* RNA abundance in infected brown Norway rat bubos versus growth in LB (46) (Fig. 5A).

The genome viewer (Fig. 5B and C) displays a value on the *y* axis (the number of mapped reads or the fold change in RNA or protein, for example) and corresponding genes on the *x* axis, allowing navigation of the genome. Depending on the data type, different information is plotted. For microarray experiments where only biological condition comparisons are available, fold change between two conditions is plotted as "relative expression data." For RNA-Seq experiments, stranded or unstranded read coverage is plotted as "absolute expression data" (Fig. 5C) and fold changes can be plotted as "relative expression data." For proteomics data, both relative and absolute (intensity; Fig. 5C) expression data can be plotted if available. A multi-omics view of different comparisons can be plotted this way, such as infected brown Norway rat bubos versus growth in LB (46), *Y. pestis* RNA abundance in infected C57BL/6 mouse lungs versus growth in BHI (44), and infected human plasma versus growth in LB (45) (Fig. 5B). A

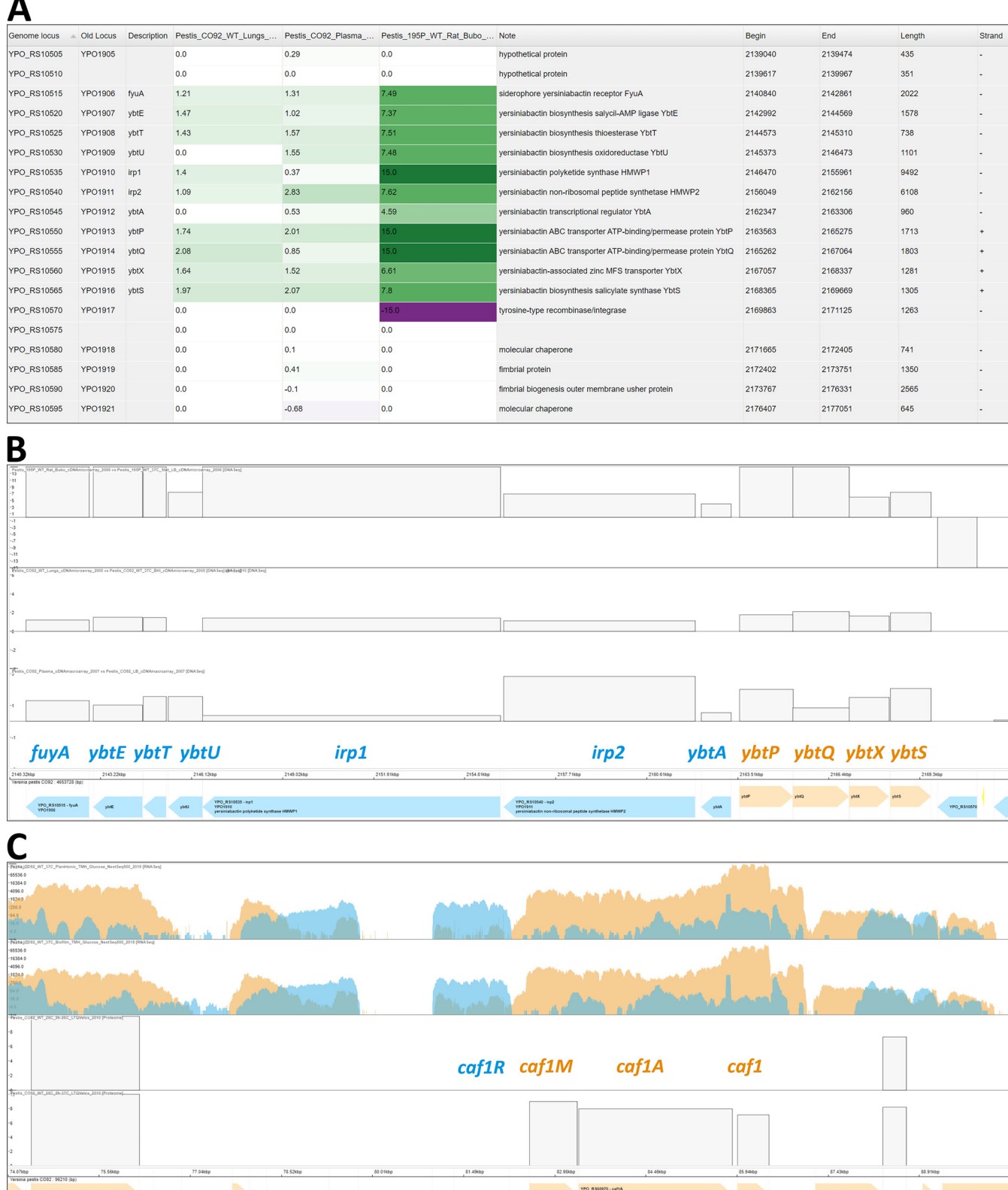

**FIG 5** (A) Heatmap viewer. Heatmap view of the yersiniabactin siderophore *ybt* operon in the CO92 strain genome, displaying three *in vivo* and *ex vivo* experiments comparing (from left to right): *Y. pestis* RNA abundance in infected C57BL/6 mouse lungs versus growth in BHI (44), *Y. pestis* RNA abundance after growth in human plasma versus growth in LB (45), and *Y. pestis* RNA abundance in infected brown Norway rat bubos versus growth in LB (46). (B) Genome viewer. Multi-omics view showing the relative expression of the yersiniabactin siderophore *ybt* operon in the CO92 genome for the following

multi-omics view of absolute expression can also be plotted on this viewer, such as the two RNA-Seq stranded coverages, 37°C growth in planktonic and biofilm states in TMH medium supplemented with glucose (47), and LC-MS/MS data showing protein abundance after 8 h growth at 26°C or 37°C (48) (Fig. 5C). A specific view can be saved and downloaded in a .gview file with the "Save data selection" button. The .gview file can later be uploaded in a new session with the "Load data selection button" or from the home page.

For the 26 RNA-Seq experiments which included replicated samples, a quality control and differential analysis report is accessible from the transcriptomic browser in the "DESeq2 report" column. This column links to an interactive web page where different charts and tables make it possible to assess the overall quality of the experiment and bioinformatic pipeline. For example, the 36 *Y. pseudotuberculosis* YPIII samples generated by Avican et al. to create an RNA atlas of human pathogens exposed to different stresses can be examined (49). Mapped read counts are heterogenous between samples, but more than 2.5 million reads could be mapped for all runs except for the run SRR11998801 (Fig. 6A). Based on the feature counts, hierarchical clustering and interactive three-dimensional (3D) principal-component analysis (Fig. 6B) showed a good clustering of sample replicates. Run SRR11998801 differed from the two other replicates of the same condition. Interactive volcano plots allow rapid exploration of the up- and downregulated genes in every computed comparison within the same experiment, such as the bile acid stress condition versus untreated condition YPIII (Fig. 6C). Sortable tables, which can also be filtered by locus name or gene name, allow further exploration of regulations, and the processed data can be downloaded in CSV format. We recommend using Mozilla Firefox to benefit from all the functionalities implemented in the RNA-Seq reports.

**(ii) The gene viewer.** From the home page, a quick access to the gene viewer for reference strains is implemented (Fig. 2, top). In the gene viewer, each gene is accessible via a gene list or a graphical view after a specific *Yersinia* genome is selected (Fig. 7). A search bar makes it possible to quickly filter the list by gene name, gene locus, or any field present in the "General information" panel as described below.

For a specific open reading frame (ORF), information such as gene locus, name, product, position, strand, and size (in base pairs and amino acids) can be found under the "General information" panel. A link to the GenBank protein and the number of genomes presenting a homolog in the Yersiniomics database is also present, in addition to the graphical view presenting the genetic environment of the gene. Other general information extracted from the "features" field of the GenBank GFF file is also displayed. A synteny based on the SynTView software (50) was also implemented for some reference genomes: *Y. pestis* CO92 and KIM, *Y. pseudotuberculosis* IP32953 and YPIII, and *Y. enterocolitica* Y11 and 8081 (Table 1). As synteny data are heavy to download and the SynTView software requires a significant amount of memory and computational power, loading is optional and conditioned to the "Show synteny" button in the "General information" panel. After the loading of SynTView, the software displays the beginning of the genome. When clicking on or searching a gene in the Yersiniomics gene viewer panel, SynTView automatically centers on the corresponding gene. The software allows the user to browse the genetic environment and conservation of each gene among other closely related genomes present in Yersiniomics. SynTView allows the user to explore syntenic organization and dynamically zoom in on a position by genomic location or go directly to specific genes by name. The view is linked with dynamic interactions with other specialized views (circular view and dot plot).

**FIG 5** Legend (Continued)

conditions (from top to bottom): infected brown Norway rat bubos versus growth in LB (46), *Y. pestis* RNA abundance in infected C57BL/6 mice lungs versus growth in BHI (44) and infected human plasma versus growth in LB (45). Induction of the yersiniabactin operon is observed both *in vivo* and *ex vivo*. (C) Genome viewer. Multi-omics view showing the absolute expression of the pseudocapsule *caf* operon in the CO92 pMT1 plasmid for the following conditions (from top to bottom): two RNA-Seq stranded coverages, growth at 37°C in planktonic and biofilm states in TMH medium supplemented with glucose (47), and LC-MS/MS data showing protein abundance after 8 h growth at 26°C or 37°C (48) where the *caf* operon is induced.

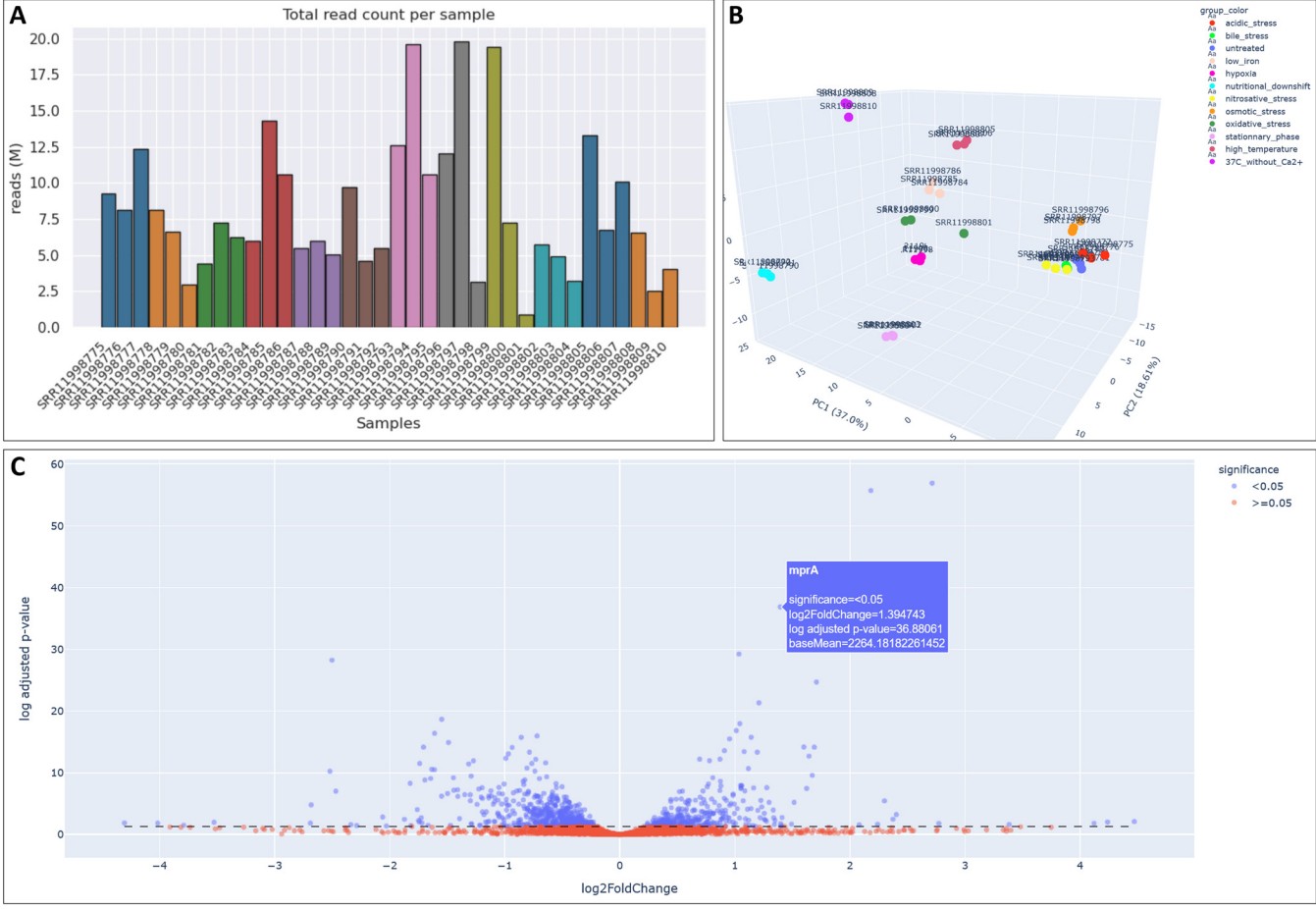

**FIG 6** DESeq2 differential analysis report. (A) Mapped read counts of the 36 *Y. pseudotuberculosis* YPIII samples exposed to different stresses generated by Avican et al. (49). (B) Interactive 3D principal-component analysis based on mapped read counts showing good clustering between replicates among the 36 samples exposed to different stresses. (C) Interactive volcano plot of the bile acid stress condition versus untreated condition for the YPIII strain, allowing the user to rapidly explore the up- and downregulated genes.

The "Homologs" panel represents the 200 genomes present in the Yersiniomics database next to the phylogenetic tree (Fig. 8). For each genome, the homologous locus, old locus, and linked GenBank protein ID are displayed, next to the BLASTP results, such as the percentage of coverage and percentage of similarity on the covered region, or the BLAST E value and bit score. The "Bidirectional" column indicates if the best BLASTP hit of the homologous gene is reciprocal. Genomes are highlighted in blue in the table when searched with the search box, and genomes selected in the table are highlighted in red on the phylogenetic tree displayed on the left. The product of the percentage of coverage times the percentage of similarity is displayed alongside the gene locus in the phylogenetic tree, which can be exported in SVG format. The corresponding gene can be opened in its own gene viewer by double-clicking on its line in the homolog table, facilitating navigation between *Yersinia* homologs and access to their respective omics data.

For genomes presenting transcriptomic data sets such as microarray or RNA-Seq data, the panel "Transcript differential expressions" displays the fold change of the gene in each available transcriptome comparison (Fig. 9). These fold changes can be filtered with a cutoff to select over- or underexpressed genes over a certain threshold. A fold change of +15 or −15 corresponds to the absence of the transcript under one of the conditions. A fold change of exactly 0.0 indicates the absence of transcript under both conditions or that the information for this gene under these conditions could not be retrieved. For most of the RNA-Seq data, if replicates were measured, the *P* value

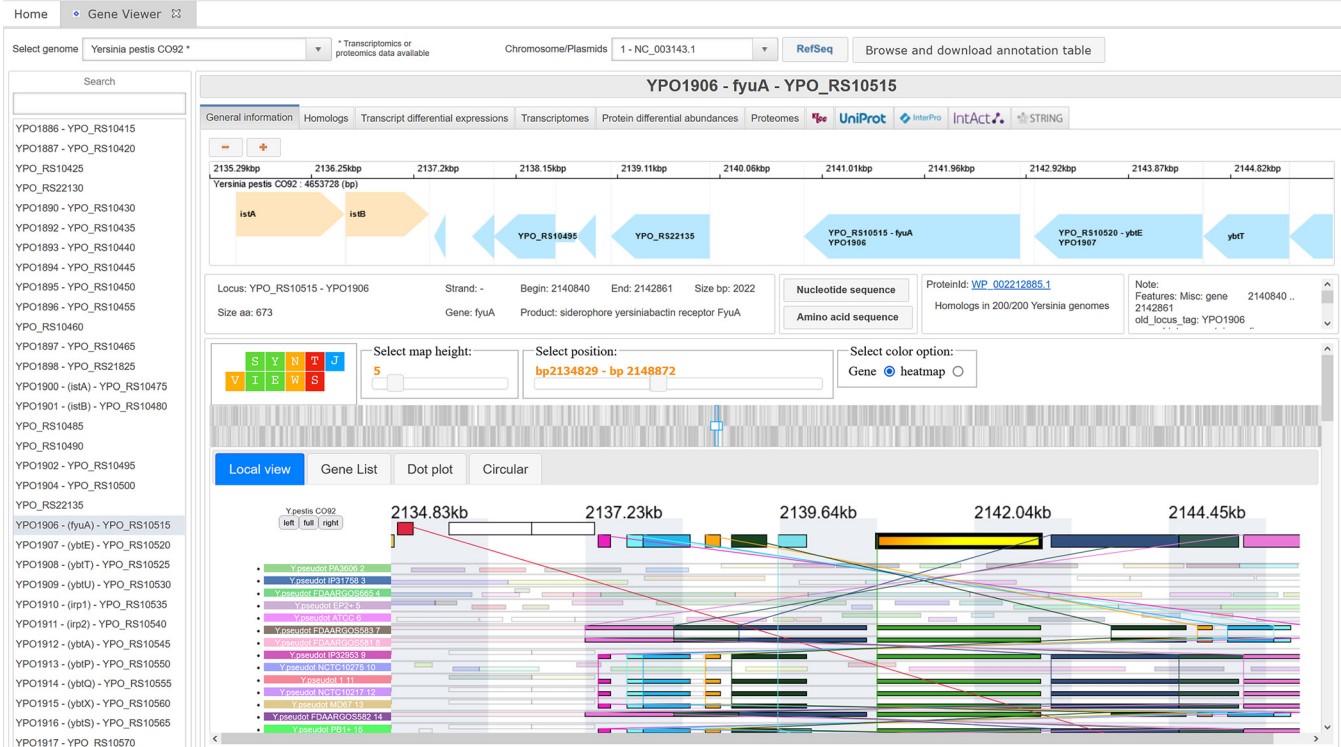

**FIG 7** The gene viewer general information panel. Information about the *fyuA*/*psn* gene of *Yersinia pestis* CO92 (YPO1906) is displayed, such as the RefSeq new and old locus names, the length of the gene and protein, the gene position in the genome, and a graphical view of the surrounding genomic region. Nucleotide and amino acid sequences can be visualized and downloaded. A search bar to find other genes in the genome is available in the left column. A switch between the chromosome and the plasmids is available in the upper banner next to the "Chromosome/Plasmids" label. For several reference strains, synteny can be shown in the SynTView software. Selection of a gene on the Yersiniomics gene viewer automatically searches for the corresponding gene in SynTView. Genomes are aligned on the searched gene, which is shaded in yellow. Homologous genes are displayed with the same color. Genes of the reference genome can also be searched in the "Gene List" tab. Genome rearrangement between the reference and another genome can be visualized in the "Dot plot" tab. The "Circular" tab allows the user to map all the genomes on the reference genome on a circular plot.

and adjusted *P* value were computed and are displayed with the comparisons. The results can be filtered using a cutoff on the *P* value. For the microarray experiments and the RNA-Seq experiments without replicates, the *P* value and adjusted *P* value were set to 0.0 and the experiments cannot be filtered using the *P* value. Each comparison can be selected and displayed in two different ways, in the genome viewer or in the heatmap viewer, as detailed above. For RNA-Seq data, an additional panel called "Transcriptomes" displays the transcripts per million (TPM) normalized value (51) of the raw read counts mapped to the gene, allowing the user to assess whether the gene is highly transcribed.

For genomes presenting semiquantitative proteomic data, a similar panel called "Protein differential expressions" displays the fold change for the abundance of the gene-encoded protein in the available comparisons, very similarly to the "Transcript differential expressions" panel. The same customizable fold change cutoff is available, as well as the genome viewer and the heatmap viewer. A protein which is detected under one condition and not the other will also have a $\log_2$(fold change) set to $+15$ or $-15$. Experiments can also be filtered using the *P* value, and adjusted *P* values are shown next to the *P* value when present or set to 0.0 when they were not computed or not retrieved from publications. To assess for the presence of a protein in a semi-quantitative or nonquantitative proteomic experiment, the "Proteomes" panel displays the biological conditions in which the protein was detected, with its associated raw value (label-free quantitation [LFQ] or FTICR intensity) when available.

For several reference strains, dynamic access to KEGG, UniProt, InterPro, IntAct, and STRING was implemented at the gene level (Table 1). When a specific gene is selected, these tabs are automatically linked to the corresponding entries in each website. This

**TABLE 1** Genomes with computed synteny and access to external databases[a]

| Strain | Synteny | KEGG, UniProt, InterPro | STRING |
|---|---|---|---|
| *Y. aldovae* 670-83 | | Yes | Yes |
| *Y. aleksiciae* 159 | | Yes | |
| *Y. alsatica* SCPM-O-B-7604 | | Yes | |
| *Y. canariae* NCTC 14382 | | Yes | |
| *Y. enterocolitica* 1055Rr | | Yes | |
| *Y. enterocolitica* 2516-87 | | Yes | |
| *Y. enterocolitica* 8081 | Yes | Yes | Yes |
| *Y. enterocolitica* FORC_002 | | Yes | |
| *Y. enterocolitica* WA | | Yes | |
| *Y. enterocolitica* Y11 | Yes | Yes | |
| *Y. enterocolitica* YE53/03 | | Yes | |
| *Y. entomophaga* MH96 | | | Yes |
| *Y. hibernica* CFS1934 | | Yes | |
| *Y. hibernica* LC20 | | Yes | Yes |
| *Y. intermedia* Y228 | | Yes | Yes |
| *Y. massiliensis* GTA | | Yes | Yes |
| *Y. mollaretii* ATCC 43969 | | Yes | Yes |
| *Y. occitanica* ATCC 33639 | | | Yes |
| *Y. occitanica* Y225 | | Yes | |
| *Y. occitanica* Y231 | | Yes | |
| *Y. pestis* 91001 | | Yes | |
| *Y. pestis* A1122 | | Yes | |
| *Y. pestis* Angola | | Yes | |
| *Y. pestis* Antiqua | | Yes | |
| *Y. pestis* CO92 | Yes | Yes | Yes |
| *Y. pestis* D106004 | | Yes | |
| *Y. pestis* D182038 | | Yes | |
| *Y. pestis* El Dorado | | Yes | |
| *Y. pestis* Harbin 35 | | Yes | |
| *Y. pestis* Harbin 35bis | | Yes | |
| *Y. pestis* KIM10+ | | Yes | Yes |
| *Y. pestis* KIM5 | Yes | Yes | Yes |
| *Y. pestis* Nepal516 | | Yes | |
| *Y. pestis* PBM19 | | Yes | |
| *Y. pestis* Pestoides F | | Yes | |
| *Y. pestis* Shasta | | Yes | |
| *Y. pestis* Z176003 | | Yes | |
| *Y. pseudotuberculosis* 1 | | Yes | |
| *Y. pseudotuberculosis* ATCC 6904 | | Yes | Yes |
| *Y. pseudotuberculosis* EP2+ | | Yes | |
| *Y. pseudotuberculosis* IP31758 | | Yes | |
| *Y. pseudotuberculosis* IP32953 | Yes | Yes | |
| *Y. pseudotuberculosis* IP32953bis | | Yes | |
| *Y. pseudotuberculosis* MD67 | | Yes | |
| *Y. pseudotuberculosis* PA3606 | | Yes | |
| *Y. pseudotuberculosis* PB1+ | | Yes | |
| *Y. pseudotuberculosis* YPIII | Yes | Yes | |
| *Y. rohdei* YRA | | Yes | Yes |
| *Y. ruckeri* Big Creek 74 | | Yes | |
| *Y. ruckeri* SC09 | | | Yes |
| *Y. ruckeri* YRB | | Yes | |
| *Y. similis* 228 | | Yes | |

[a]Species nomenclature proposed by the French *Yersinia* National Reference Center.

allows very quick access to diverse information, such as the implication in biological pathways in KEGG (Fig. 10A), the Alphafold prediction structure in UniProt (Fig. 10C), the protein domains and Pfam families in InterPro, protein-interacting partners identified with methods such as a yeast two-hybrid screen against human proteins (52) accessible in IntAct (Fig. 10D), or other metalinks such as co-occurrence in abstract computed in the STRING database (Fig. 10B). A button allows the user to directly open the corresponding website in a new tab of the web browser. Due to cookie policies, we recommend using Mozilla Firefox to access the STRING tab embedded in Yersiniomics.

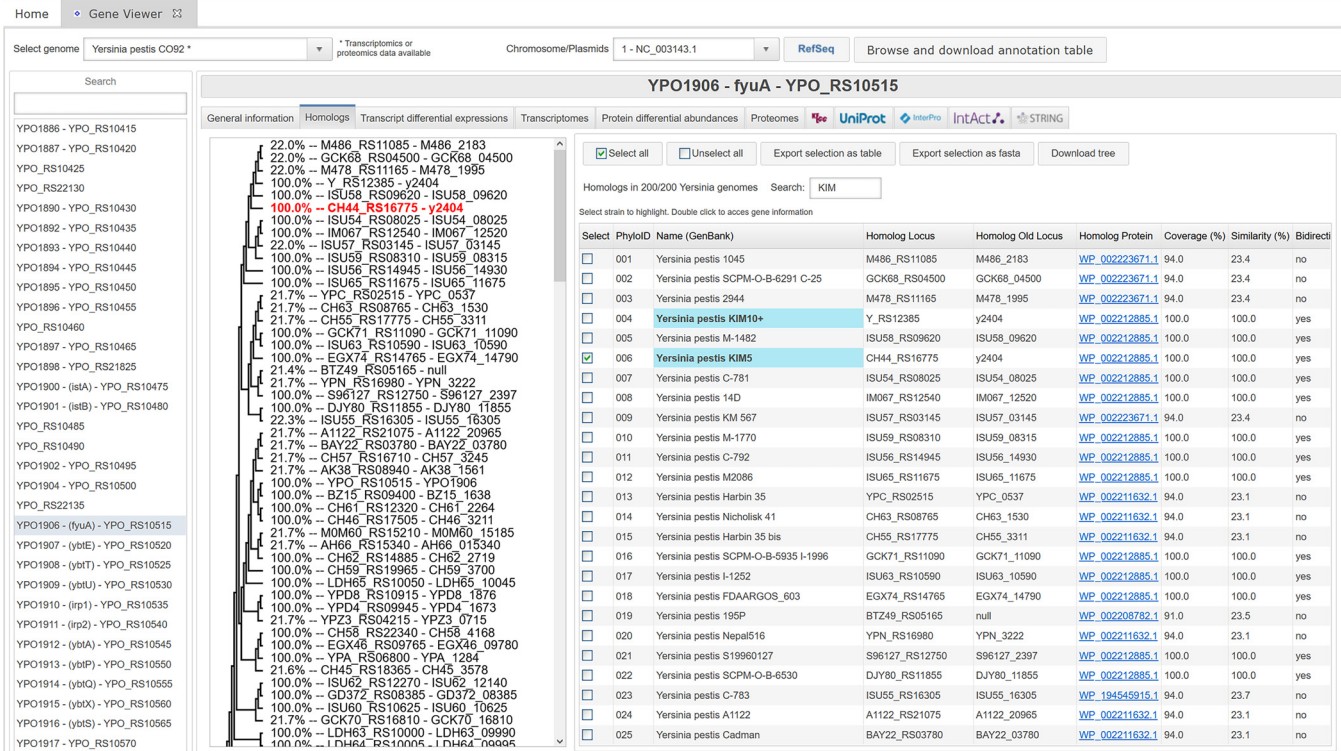

**FIG 8** The gene viewer homolog panel. The table of the *fyuA* homologs in the Yersiniomics database is displayed on the right, with the gene locus, the BLASTP results (such as percent coverage and percent similarity on this coverage), the E value and bit score, and the name of the corresponding strain. The product of the percentage of similarity times the percent coverage, as well as the name of the homologous gene, are also displayed on the phylogenetic tree on the left and highlighted when selected. A multi-fasta protein file of the selected homologs can be downloaded.

**Data loading.** From the home page, two buttons allow the user to load a .gview file previously saved from the genome viewer and to download transcriptomics and proteomics processed data (Fig. 2, bottom panel). These data are sorted in the "Transcriptomes" and "Proteomes" directories, each one subdivided into strain-specific directories. Excel files showing genes in rows and biological conditions or comparisons of the selected strain in columns are available to download in each directory, i.e., $\log_2$(fold change) table (Table_LOGFC_strain_name.excel), associated *P* value (Table_PVALUE_strain_name.excel), and associated adjusted *P* value (Table_PADJ_strain_name.excel) for the comparisons, TPM-normalized counts for RNA-Seq experiments (AllRNA-SeqTPM_strain_name.excel), and intensity values for proteomes (Table_Expr_strain_name.excel). The TPM-normalized count can then be used by each user to compare gene expression within the same sample or compute coexpression networks.

**Genomic data set description.** We collected 200 assemblies of *Yersinia* genomes, among which 10 were sequenced twice (annotated "bis") by different laboratories with different technologies (such as Sanger or combination of Illumina and Nanopore or PacBio sequencing). Of the 190 unique genomes, we gathered 61 *Y. pestis*, 24 *Y. pseudotuberculosis*, and 37 *Y. enterocolitica* genomes. In addition, 37 strains of the fish pathogen *Y. ruckeri* were also collected, as well as one complete genome of the insect pathogen *Y. entomophaga* and 30 genomes of nonpathogenic *Yersinia* (Table 2).

In the genomics browser, the "Name (GenBank)" column refers to the GenBank taxonomic assignation, but particular attention should be drawn to the "Most recent assignation (cgMLST)" column, where 9 strains were reassigned by the cgMLST scheme developed in our laboratory (40).

Of note, the RefSeq-reannotated locus name, recognizable by its "..._RS..." pattern, was used to access the genes differing from the initial widely used locus name, such as "YPO..." for *Y. pestis* CO92 or "YPTB..." for *Y. pseudotuberculosis* IP32953. We added this previous name in a column named "Old Locus" in the heatmaps and this old

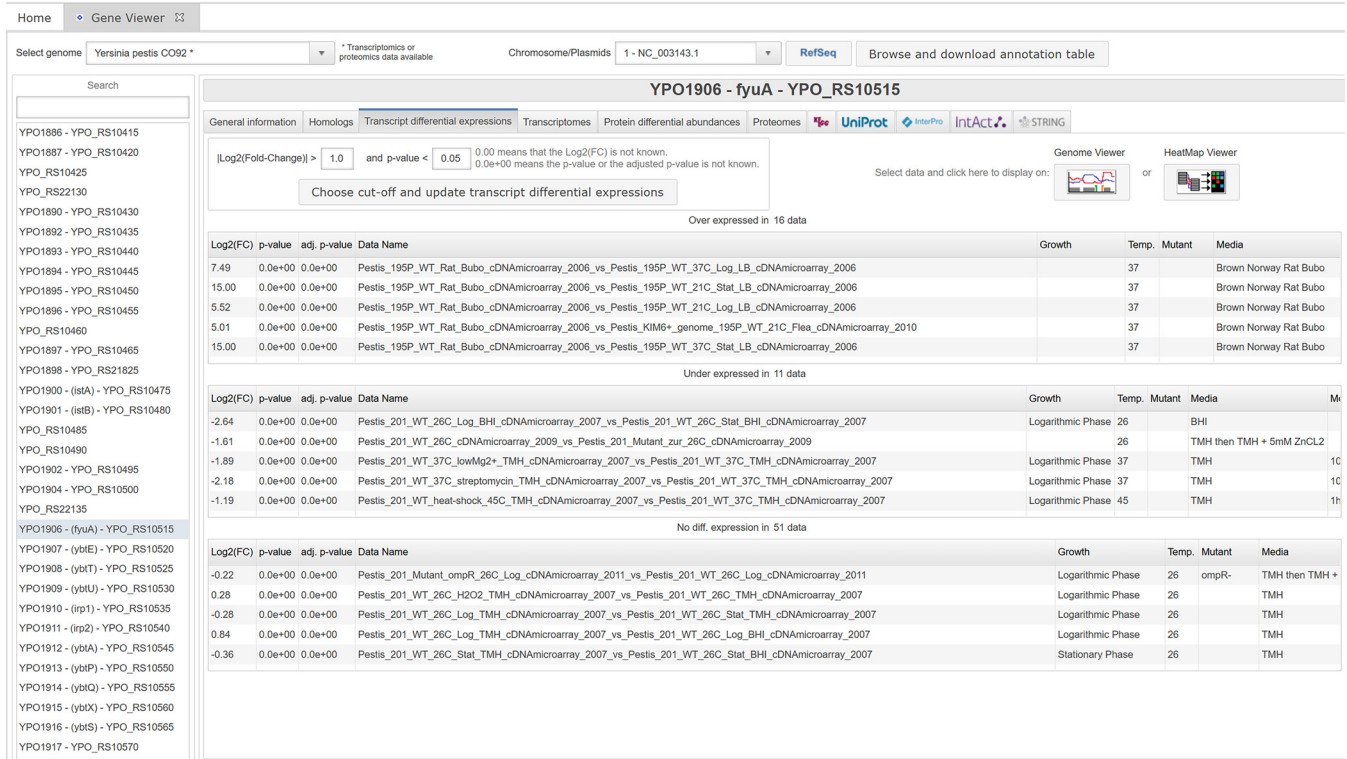

**FIG 9** The gene viewer transcript differential expression panel. After selection of a log$_2$(fold change) cutoff, comparison of biological conditions is dispatched in three panels. For RNA-Seq experiments, results are also filtered by selecting a *P* value cutoff. The top panel shows upregulated conditions and corresponding fold change, *P* value, and adjusted *P* value. For example, *fyuA* expression is upregulated 180-fold in the rat compared to growth in LB at 37°C and upregulated 46-fold compared to growth in LB at 21°C. The middle panel shows downregulated conditions and the corresponding fold change. Expression of *fyuA* is decreased 6.2-fold in exponential phase compared to stationary phase in BHI at 26°C and decreased 4.5-fold in the presence of streptomycin at 37°C compared to the untreated condition. The lower panel displays condition comparisons in which fold change is below the selected cutoff and *P* value, when available. The conditions can be displayed with the genome viewer or the heatmap viewer via the two buttons on the top right.

locus is present after the RefSeq locus name in the gene list of a genome, separated by a dash, and followed by the gene name in parentheses. For example, the *mioC* gene in the CO92 genome can be found as "YPO_RS01005 - YPO0001 (mioC)."

The *Y. pestis* community widely uses the CO92 and KIM locus names to refer to genes, using the formats "YPO..." and "y...," respectively. However, the KIM "y..." locus can be found only in the KIM10+ genome annotation in GenBank, a strain lacking several plasmids (pCD1/pYV and pPCP1/pPla) compared to its parental strain, KIM5. To map the RNA-Seq data to the most complete genome but conserve the known locus numbers widely used by the *Y. pestis* community, we reannotated the old locus of the KIM5 genome with the KIM10+ old locus using a systematic BLAST search between both genomes' open reading frames.

Of note, the YPIII genome in GenBank lacks the pIB1 plasmid (the YPIII pYV virulence plasmid coding for a type III secretion system). Thus, this plasmid was not included in the database.

**Transcriptomics data set description.** Transcriptomic experiments were collected for *Yersinia* reference strains such as *Y. pestis* CO92, KIM, 91001, and Pestoides F, *Y. pseudotuberculosis* YPIII and IP32953, *Y. enterocolitica* 8081 and Y11, *Y. ruckeri* CSF007-82, and *Y. entomophaga* MH96 (Table 3). Of note, some strains' data are mapped on genomes of other strains, such as *Y. pestis* 201 Microtus, whose microarray data were generated using CO92 microarray, or RNA-Seq data that were mapped to strain 91001 Microtus strain, as the strain 201 sequence is assembled only at the scaffold level. This is also the case for *Y. ruckeri* CSF007-82, whose complete assembled genomic sequence is not available, and RNA-Seq data were mapped to its genetically closest sequenced strain, QMA0440. We gathered 151 biological

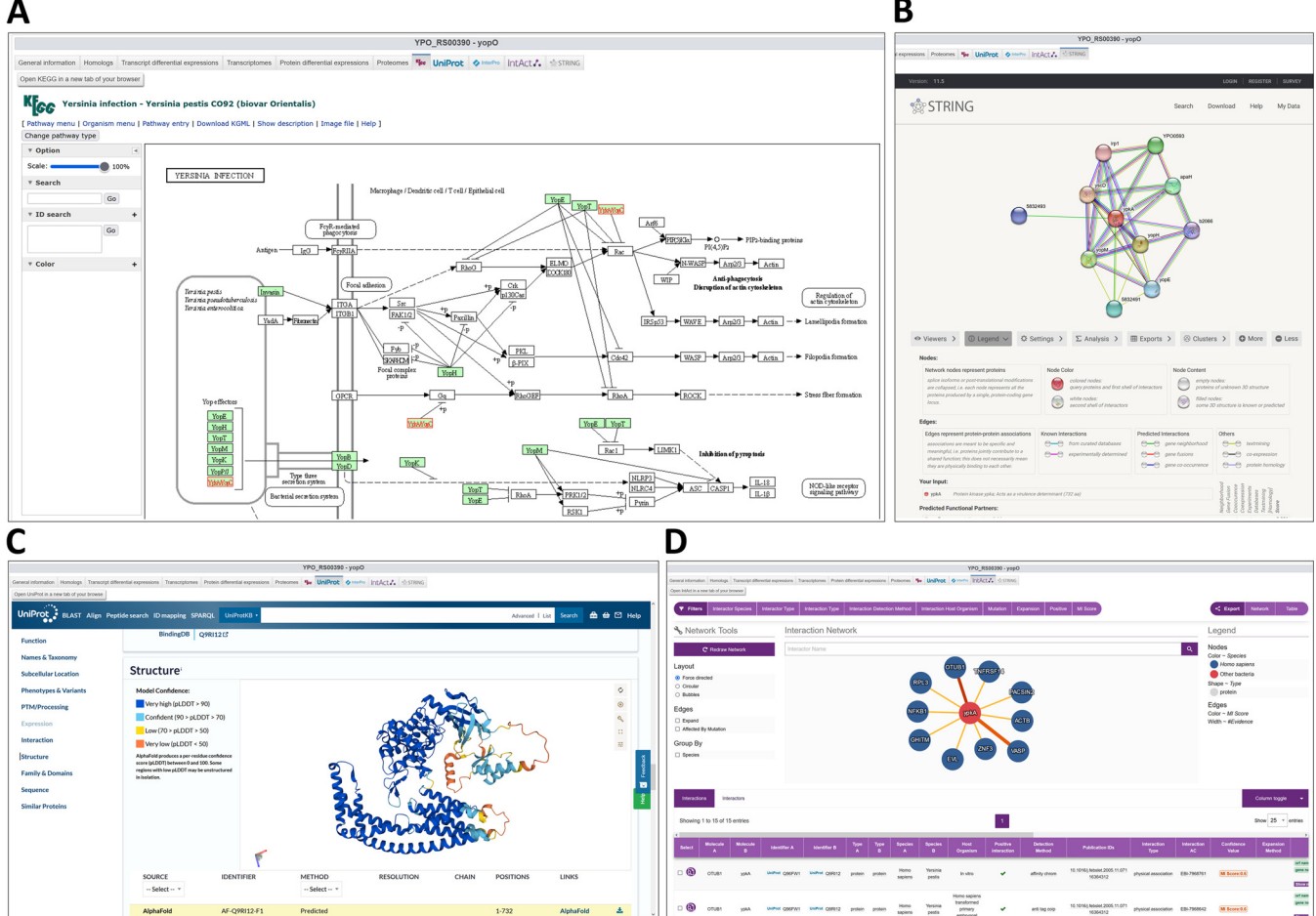

**FIG 10** Quick access to integrated views for the *yopO/ypkA* effector gene in different databases. (A) The KEGG website allows the user to browse biochemical pathways such as regulation of host actin cytoskeleton by the YpkA protein and other *Yersinia* effectors. (B) The STRING website allows the user to browse links between proteins via PubMed abstract text mining, co-occurrence across genomes, direct chemical interactions, or gene neighborhood. (C) The UniProt website allows the user to access protein annotations and Gene Ontology (GO) terms (molecular function, biological process, and cellular component), protein domains, and crystal structure deposited in PDB or Alphafold structure prediction. (D) The IntAct website allows the user to browse the interactome database and access results of pulldown or yeast two-hybrid experiment, showing interaction of YpkA with immune proteins.

conditions analyzed with microarrays and 425 RNA-Seq runs coming from 32 projects, accounting for 166 biological conditions when aggregating replicates.

Most of the microarray data (100 conditions) were mapped to the *Y. pestis* CO92 genome, as it was the first *Y. pestis* genome fully sequenced and the first available microarray. Several experiments with strains such as *Y. pestis* 195P, 201, GB, and KIM and even *Y. pseudotuberculosis* IP32953 and YPIII used the CO92 microarray; thus, subsequent data are available on the CO92 genome on Yersiniomics. A total of 9 biological conditions were mapped on the *Y. pestis* KIM genome, 8 on *Y. pestis* Pestoides F, 20 on *Y. pseudotuberculosis* YPIII (encompassing 8 IP32953 experiments), 8 on *Y. pseudotuberculosis* PB1+, and 6 on *Y. enterocolitica* 8081.

Of the 166 RNA-Seq biological conditions, 44 were mapped on the *Y. pseudotuberculosis* YPIII genome. A total of 30 come from the *Y. pestis* 201 Microtus strain and were mapped to the close relative strain 91001. Additionally, 24 biological conditions were mapped to the *Y. enterocolitica* 8081 genome, 17 were mapped to the *Y. pseudotuberculosis* IP32953 genome, 11 to the *Y. enterocolitica* Y11 genome, 9 to the *Y. pestis* CO92 genome, 8 to the *Y. enterocolitica* Y1 genome (of which 4 come from strain 8081), 8 to the *Y. entomophaga* MH96 genome, 7 to the *Y. pestis* KIM5 genome, 4 to the *Y. ruckeri* SC09 genome, and 2 to the vaccine strain *Y. pestis* EV76 genome, and 2 biological conditions with the *Y. ruckeri* strain CSF007-82 were mapped to its close relative QMA0440.

**TABLE 2** Numbers of complete *Yersinia* genomes[a]

| *Yersinia* species | No. of genomes | |
|---|---|---|
| | Unique complete | Sequenced twice |
| *Y. pestis* | 61 | 6 |
| *Y. enterocolitica* | 37 | 1 |
| *Y. ruckeri* | 37 | |
| *Y. pseudotuberculosis* | 24 | 3 |
| *Y. intermedia* | 8 | |
| *Y. occitanica* | 5 | |
| *Y. massiliensis* | 4 | |
| *Y. bercovieri* | 2 | |
| *Y. hibernica* | 2 | |
| *Y. aldovae* | 1 | |
| *Y. aleksiciae* | 1 | |
| *Y. alsatica* | 1 | |
| *Y. canariae* | 1 | |
| *Y. entomophaga* | 1 | |
| *Y. frederiksenii* | 1 | |
| *Y. kristensenii* | 1 | |
| *Y. mollaretii* | 1 | |
| *Y. rohdei* | 1 | |
| *Y. similis* | 1 | |
| Total | 190 | 10 |

[a]Species nomenclature proposed by the French *Yersinia* National Reference Center.

A summary of the 154 *Y. pestis* experiments can be found aggregated by culture temperature and strain (Table 4), culture medium and strain (Table 5), and genetic status and strain (Table 6).

**Proteomic data set description.** Many *Y. pestis* proteomes have been aggregated and processed by the Pacific Northwest National Laboratory (PNNL) to create a tool that differentiates naturally occurring and laboratory strains of *Y. pestis* (53), encompassing data from different studies (48, 54, 55) as well as PNNL archives (56). These processed intensity data sets were kindly shared by Eric D. Merkley. These data represent most of the 32 biological conditions which used whole-cell lysates of *Y. pestis* reference strain CO92 and were analyzed via shotgun LC-MS/MS using linear ion trap technologies and/or Orbitrap mass spectrometry for identification (Thermo Fisher LTQ XL/LTQ Velos). Proteomes also include 5 biological conditions from one study investigating *Y. pestis* Microtus strain 201 using Orbitrap LC-MS/MS technology (Thermo

**TABLE 3** Number of omics biological conditions mapped to reference strains

| Reference strain | No. of conditions | | |
|---|---|---|---|
| | Microarray | RNA-Seq | Mass spectrometry |
| *Y. pestis* CO92 | 100 | 9 | 32 |
| *Y. pestis* KIM | 9 | 7 | 24 |
| *Y. pestis* Microtus 91001 | | 30 | 6 |
| *Y. pestis* Pestoides F | 8 | | |
| *Y. pestis* EV76 | | 2 | |
| *Y. pseudotuberculosis* YPIII | 20 | 44 | |
| *Y. pseudotuberculosis* IP32953 | | 17 | |
| *Y. pseudotuberculosis* PB1+ | 8 | | |
| *Y. enterocolitica* 8081 | 6 | 24 | |
| *Y. enterocolitica* Y11 | | 11 | |
| *Y. enterocolitica* Y1 | | 8 | |
| *Y. ruckeri* SC09 | | 4 | |
| *Y. ruckeri* QMA0440 | | 2 | |
| *Y. entomophaga* MH96 | | 8 | |
| Total | 151 | 166 | 62 |

**TABLE 4** Number of *Y. pestis* transcriptomic experiments according to culture temperature and strain

| Culture temp (°C) | *Y. pestis* strain used | No. of expts | |
|---|---|---|---|
| | | Array | RNA-Seq |
| 10 | 201 | 1 | |
| 21 | 195P | 2 | |
| | EV76 | | 1 |
| | KIM6+ | 5 | |
| 26 | 201 | 18 | 21 |
| | CO92 | 4 | |
| | Pestoides F | 4 | |
| 28/37 | CO92 | 4 | |
| 28 | CO92 | 1 | |
| | KIM5 | 4 | |
| | KIM53 | 5 | 1 |
| 30 | CO92 | 14 | |
| 37 | 195P | 3 | |
| | 201 | 16 | 8 |
| | CO92 | 18 | 9 |
| | EV76 | | 1 |
| | GB | 2 | |
| | KIM53 | | 4 |
| | KIM6+ | | 2 |
| | Pestoides F | 4 | |
| 45 | 201 | 1 | |
| Multiple | 201 | | 1 |

Fisher QExactive), the first and only study defining the *Y. pestis* secretome (57). In addition, 24 biological conditions from 2 studies focus on the *Y. pestis* strain KIM (12 in FTICR, 12 in 2D gel electrophoresis plus LC-MS/MS) and used cell fractionation to determine protein localization (58, 59) (Table 3).

**Data set exploration and validation.** One dual RNA-Seq experiment providing gene expression profiles in a murine pneumonic plague model (60) did generate RNA-Seq raw data; however, gene content and differential expression were not analyzed. Our systematic processing pipeline allowed us to process and validate these data.

The *in vivo* experiment comparing transcripts of the *Y. pestis* KIM strain in OF1-infected mouse lungs versus HIB cultures confirmed most previous results described in the seminal microarray experiment in a pneumonic plague model that used the CO92 strain infecting C57BL/6 mice or grown in BHI (44): the methionine biosynthesis pathway genes *metA*, *metE*, *metF*, *metK*, and *metR* were among the most upregulated genes in the lungs in both studies. Similarly, the yersiniabactin operon (y2404 to y2394 in the KIM strain) was the most upregulated operon at 1 h, 24 h, and 48 h postinfection, highlighting the importance of metal acquisition by bacteria in their mammalian host. The pH 6 antigen *psaA* gene, encoding fimbriae required for virulence, was downregulated 48 h postinfection in both studies but was upregulated 1 h postinfection in the latter experiment. The cold shock-responsive gene *cspD* and genes involved in the detoxification of reactive oxygen species, such as *katA* and *katG* (*katY*), encoding catalases, and *sodB*, encoding superoxide dismutase, were downregulated in both studies, and nitric oxide-induced *hmpA* was even more upregulated in the experiment using the KIM strain than in the experiment based on the CO92 strain. However, the plasminogen activator *pla* was downregulated 24 h postinfection and not at 48 h postinfection in the study using the KIM strain, in contrast to the study using the CO92 strain. Similarly, *cspA1* and *cspA2* were upregulated in the lungs infected with the CO92 strain but downregulated in the study by Israeli et al. (60). Discrepancies between these results could be explained by differences in *Y. pestis* strains and mouse lines as well as culture conditions of bacteria *in vitro*.

**TABLE 5** Number of *Y. pestis* transcriptomic experiments according to culture medium and strain

| Culture medium[a] | *Y. pestis* strain used | No. of expts | |
|---|---|---|---|
| | | Array | RNA-Seq |
| BAB broth | GB | 2 | |
| BHI | 201 | 2 | 9 |
| | CO92 | 7 | 1 |
| Custom | CO92 | 8 | |
| | Pestoides F | 8 | |
| DMEM | KIM5 | 1 | |
| HIB | CO92 | 19 | |
| | KIM53 | | 2 |
| | KIM6+ | | 2 |
| Human plasma | CO92 | 5 | |
| LB | 195P | 4 | |
| | 201 | | 2 |
| | CO92 | 1 | |
| | KIM6+ | 3 | |
| Macrophages | CO92 | | 2 |
| | KIM5 | 3 | |
| MHB | KIM53 | 5 | |
| TMH | 201 | 34 | 11 |
| | CO92 | | 6 |
| *In vivo* | 195P | 1 | |
| | 201 | | 2 |
| | CO92 | 1 | |
| | KIM53 | | 3 |
| | KIM6+ | 2 | |
| Unknown | 201 | | 6 |
| | EV76 | | 2 |

[a]BAB, blood agar base; DMEM, Dulbecco's modified Eagle medium; MHB, Mueller-Hinton broth.

## DISCUSSION

Over the last 2 decades, the omics revolution has generated an impressive number of data sets which are often difficult to analyze in depth without any prior bioinformatic knowledge. In addition, only a fraction of generated and processed data is generally accessible and/or used in research articles. Here, we aimed at exploiting *Yersinia* omics data published over the last 20 years to make them accessible in a user-friendly way to biologists without familiarity with deep bioinformatics. We thus constructed and processed a database gathering 200 genomic, 317 transcriptomic, and 62 proteomic data sets of *Yersinia* species, which are browsable at the gene level and according to experimental conditions on the custom-made Yersiniomics website (https://yersiniomics.pasteur.fr/). Notably, three previous studies (60–62) did not measure differential gene expression under their biological conditions. These data were thus processed and included in Yersiniomics. One of these studies confirmed for the most part previous results obtained using a similar microarray approach. Other recent RNA-Seq studies did not make their raw sequencing data publicly available and could not be included in our database for now (63–66). To expand information on genes of interest, dynamic links to external databases were implemented for several reference strains, allowing the user to access KEGG pathways, UniProt annotation, InterPro domains, IntAct interactors and STRING networks at the gene level. These links also facilitate the access to other data, such as GO terms and Alphafold structure predictions, implemented in UniProt.

As new data and data types continue to be published, we intend to update and upgrade the Yersiniomics database with newly published RNA-Seq and LC-MS/MS

**TABLE 6** Number of *Y. pestis* transcriptomic experiment according to strain and genetic status

| Strain used | Genetic status[a] | No. of expts | |
|---|---|---|---|
| | | Array | RNA-Seq |
| 195P | WT | 5 | |
| 201 | *cobB* | | 1 |
| | *fur* | 2 | |
| | *fyuA* | | 2 |
| | *fyuA* deletion GCA | | 2 |
| | *hfq* pHfq | | 1 |
| | *hfq* pHfq-FLAG | | 2 |
| | *lcrG* | 1 | |
| | *ompR* | 3 | |
| | *oxyR* | 1 | |
| | pHfq-FLAG | | 1 |
| | *phoP* | 2 | |
| | *rcsB* | | 2 |
| | *rcsD*$_{pestis}$::*rcsD*$_{pseudotb}$ | | 2 |
| | WT pFLAG | | 3 |
| | *yfiQ* | | 1 |
| | *zur* | 1 | |
| | WT | 26 | 13 |
| CO92 | *crp* pCD1$^-$ | | 2 |
| | pCD1$^-$ | | 4 |
| | *pgm* | 15 | |
| | *pgm luxS* | 2 | |
| | *pgm ypeIR* | 2 | |
| | *pgm ypeIR yspIR* | 1 | |
| | *pgm ypeIR yspIR luxS* | 2 | |
| | *pgm yspI* | 2 | |
| | *pgm yspI ypeIR* | 1 | |
| | pPCP1$^-$ | 8 | |
| | WT | 8 | 3 |
| EV76-CN | WT | | 2 |
| GB | *dam* | 1 | |
| | WT | 1 | |
| KIM5 | WT | 4 | |
| KIM53 | WT | 5 | 5 |
| KIM6+ | pCD1$^-$ | 5 | |
| | *psaE* | | 1 |
| | WT | | 1 |
| Pestoides F | WT | 8 | |

[a]WT, wild-type strain; mutant strain (*mutated locus*).

data, with omics data generated in our laboratory, and with already available data types such as small RNAs (61–68), transcriptional start sites (TSS) (67–69), and riboswitches (67, 69). New analyses, such as gene ranking using TPM counts for each RNA-Seq experiments, will also be implemented. Ultimately, mutant phenotypes associated with specific gene locus could also be added based on signature-tagged mutagenesis screening (70–75), high-throughput transposon site hybridization procedure (76), or more recent next-generation sequencing using techniques such as transposon-insertion sequencing or transposon-directed insertion sequencing (77–81).

## MATERIALS AND METHODS

**Genomic data collection.** Genomic data were browsed on the NCBI database with the keyword "Yersinia" (https://www.ncbi.nlm.nih.gov/data-hub/genome). Assemblies were filtered at the "chromosome" or "complete" level, and "chromosome" assemblies were manually curated to verify data completeness. The most recent RefSeq annotated assemblies were downloaded in December 2022.

**Transcriptomic data collection.** Microarray data were collected on Gene Expression Omnibus (https://www.ncbi.nlm.nih.gov/geo/), and the differential expressions calculated by data depositors were directly used when available. Alternatively, tables and supplemental tables were downloaded from articles and formatted. RNA-Seq data sets were browsed on the European Nucleotide Archive (https://www.ebi.ac.uk/ena/) using the keywords "Yersinia" and "transcriptome," "RNA-Seq," or "RNASeq" and on the Sequence Read Archive (https://www.ncbi.nlm.nih.gov/sra/) using the keyword "Yersinia" and selecting the "RNA" source. Raw data, consisting of sequencing reads in fastq format, were downloaded from the SRA FTP server http://ftp.sra.ebi.ac.uk/vol1/ via the ENA website.

**Proteomic data collection.** As the treatment of LC-MS/MS raw data from the PRIDE repository is more complex and highly dependent on the mass spectrometry technology used, we directly retrieved the calculated differential expression from the tables and supplemental tables published in the literature when available. Alternatively, processed intensity data were kindly provided by their authors when data were not available online.

**Genomic data processing.** Genomes were directly processed through the Bacnet platform (33) and implemented in Yersiniomics. From the collected genomes, a phylogenetic tree was reconstructed using the 500 genes of our *Yersinia* cgMLST scheme recently developed in our laboratory (40). The 500 genes were concatenated and aligned using MAFFT v7.453 (82). Phylogenetic reconstruction was performed using IQTREE v2.0.6 (83), and the tree was rooted on *Yersinia entomophaga* MH96 (84), as it is the most ancestral branch of the genus *Yersinia* (40). The tree was drawn using Iroki (85). The taxonomic classification of the 66 *Y. pestis* genomes was based on the presence of SNPs defining the lineages, following the nomenclature developed in previous works (41, 42). *Y. pseudotuberculosis* IP32953 genome was also included in the analysis, to recover the ancestral genotypes. Briefly, either short-read sequencing data or contigs were mapped onto the *Y. pestis* CO92 reference genome (NC_003143) using the Snippy pipeline with default parameters (https://github.com/tseemann/snippy) to identify variants in the chromosome. Identified variants were inspected, and those falling within repetitive sequences (i.e., IS) were excluded, while variants associated with putative recombination events were identified using Gubbins v3.2.0 (86) and filtered out. The final set of variants ($n = 14,273$) was then used to reconstruct a maximum-likelihood phylogeny using IQ-TREE 2 (83) to classify the different *Y. pestis* genomes present in Yersiniomics into known evolutionary branches (0.PE2, 0.PE4, 0.PE3, 0.ANT, 1.ORI, 1.IN, 1.ANT, 2.ANT, 2.MED, and 4.ANT). For the other species, lineages were assessed from cgMLST.

**Homolog database and synteny.** A protein BLAST search was automatically performed for each gene of each genome against the other genomes to construct the homolog database, using the BLAST+ command-line tool v2.13.0 (87). Synteny was constructed using a best-hit bidirectional BLAST search and implemented in the SynTView framework (50). The tool was reprogrammed from Flash to JavaScript with the HTML5 2d Canvas JavaScript library Konva (https://konvajs.org/), which enables high-performance animations. Drawing strategies were developed to ensure that graphical outputs remain below the technological limits of drawing more than 10,000 dynamic graphical objects, using a mix of static and dynamic objects. SynTViewJS can be used on a public web application at https://plechat.pages.pasteur.fr/syntviewjs/. The code repository is at https://gitlab.pasteur.fr/plechat/syntviewjs.

**Transcriptomic data processing.** Downloaded microarray fold change tables were processed through the Bacnet platform. RNA-Seq raw data were processed using Sequana v0.14.6, Sequana-rnaseq v0.17.0, and Sequana-pipetools v0.10.0s (88), implementing the following software with default parameters: reads were trimmed using fastp v0.20.1 (89), processed reads were then aligned on the selected reference genome with Bowtie2 v2.4.4 (90), and strandedness was automatically identified by the Sequana pipeline. Mapped reads were then quantified using featureCounts (package subread v2.0.1) (91), and differential analysis was performed on raw read counts using DESeq2 v1.38.2 with R v4.2.1 with default parameters and Cook's cutoff enabled (43). *P* values and adjusted *P* values were calculated using default DESeq2 parameters consisting, respectively, of the Wald test and the Benjamini-Hochberg correction. Fold change and adjusted *P* value tables were extracted from DESeq2 results and processed using the Bacnet platform. BAM files generated by the Sequana pipeline were converted to strand-specific .wig files using the strand_cov function from the stranded-coverage package (https://github.com/pmenzel/stranded-coverage.git) and then processed using the Bacnet platform. Raw read counts from featureCounts were also normalized using the TPM method (51) to be displayed in the transcriptome panel of the gene viewer.

**Proteomic data processing.** For the differential analyses of one condition versus another, proteins identified in the reverse and contaminant databases and proteins "only identified by site" were first discarded from the list of identified proteins. Then, only proteins with at least three quantified intensities in a condition were kept. Differential expression was performed using LFQ when at least 1 unique peptide was detected and at least 2 peptides were quantified in a condition. A normalization was applied within the same condition centered on means of the medians (92). Remaining proteins without any intensity value in one of two conditions were considered quantitatively present in a condition and absent in another. They were therefore set aside and considered differentially abundant proteins. Missing values were imputed using the SLSA method thanks to the R package imp4p (93). Differential analysis was performed with a limma *t* test and an adaptative Benjamini-Hochberg correction to adjust the *P* values with the cp4P R package (94). Proteins with a fold change less than 2.0 were considered not significantly differentially abundant. Statistical testing of the remaining proteins (having a fold change greater than 2.0) was conducted using the limma *t* test (95).

These calculated fold changes were formatted and processed using the Bacnet platform, in parallel to fold change tables downloaded from published articles.

**Yersiniomics website design.** Yersiniomics was constructed with the Bacnet platform (33), based on Java and Eclipse e4 RCP/RAP API, and we contributed to its most recent update.

**Links to external databases.** The link to the KEGG database website uses the KEGG species identifier ("ype" for *Y. pestis* CO92) and the old locus name (for instance, "YPO0001" for the *mioC* gene of *Y. pestis* CO92), dynamically queried by URL in the form "https://www.genome.jp/entry/ype:YPO0001." UniProt accessions are dynamically retrieved and parsed with the KEGG species identifiers and the old locus name using KEGG REST Application Programming Interface (API), queried by URL in the form "https://rest.kegg.jp/conv/uniprot/ype:YPO0001." When a UniProt accession is retrieved ("A0A0H2W280" for the "YPO0001" old locus name of *Y. pestis* CO92), UniProt and InterPro are dynamically queried by URL in the forms "https://www.uniprot.org/uniprotkb/A0A0H2W280" and "https://www.ebi.ac.uk/interpro/protein/UniProt/A0A0H2W280," respectively. IntAct is dynamically accessed with the old locus name by URL in the form "https://www.ebi.ac.uk/intact/search?query=YPO0001" or "https://www.ebi.ac.uk/intact/search?query=YPO0001%20A0A0H2W280" if a UniProt accession was retrieved. A STRING URL is dynamically retrieved with the STRING species identifier ("214092" for *Y. pestis* CO92) and the old locus name, querying the STRING API in the form "https://version-11-5.string-db.org/cgi/network?taskId=bI0qgHbfGxqi&sessionId=b41GkPqd7zjt." The retrieved URL is then accessed.

**Data availability.** The source code of the Yersiniomics website, based on the Bacnet platform, is available on the GitHub repository https://github.com/becavin-lab/bacnet/. All processed data can be directly downloaded from the Yersiniomics website.

## ACKNOWLEDGMENTS

The project received funding from Institut Pasteur, Agence de l'Innovation de Défense (AID-DGA), Université Paris Cité, CNRS, LabEX Integrative Biology of Emerging Infectious Diseases (ANR-10-LBX-62-IBEID), Fondation pour la Recherche Médicale (FDT202204015222), and the Inception program (Investissement d'Avenir grant ANR-16-CONV-0005). The funders had no role in study design, data collection and interpretation, or the decision to submit the work for publication.

We thank Eric D. Merkley for sharing Pacific Northwest National Laboratory data sets. We are grateful to all members of the *Yersinia* research unit and the French national reference center for plague and other yersiniosis for insightful discussions.

We declare no conflict of interest.

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
