## [Reviewer comments · Microbiology Spectrum]

Microbiology Spectrum

Yersiniomics, a multi-omics interactive database for *Yersinia* species

Pierre Lê-Bury, Karen Druart, Cyril Savin, Pierre LECHAT, Guillem Mas Fiol, Mariette Matondo, Christophe Becavin, Olivier Dussurget, and Javier Pizarro-Cerdá

Corresponding Author(s): Javier Pizarro-Cerdá, Institut Pasteur

Review Timeline:

Submission Date:	October 17, 2022
Editorial Decision:	November 20, 2022
Revision Received:	January 9, 2023
Accepted:	January 26, 2023

Editor: Tino Polen

Reviewer(s): Disclosure of reviewer identity is with reference to reviewer comments included in decision letter(s). The following individuals involved in review of your submission have agreed to reveal their identity: Kemal Avican (Reviewer #2)

Transaction Report:

DOI: <https://doi.org/10.1128/spectrum.03826-22>

November 20, 2022

Dr. Javier Pizarro-Cerdá
Institut Pasteur
Unité de Recherche Yersinia
Paris
France

Re: Spectrum03826-22 (Yersiniomics, a multi-omics interactive database for *Yersinia* species)

Dear Dr. Javier Pizarro-Cerdá:

thank you for submitting your manuscript to Microbiology Spectrum.

I have received comments on your manuscript from two experts. Both very much acknowledge your efforts and suggested modifications for clarifications and improvements. I do hope you will find the reviewers' comments below helpful and look forward to receiving a revised version from you.

Link Not Available

Sincerely,

Tino Polen
Editor, Microbiology Spectrum

Journals Department
Reviewer comments:

Reviewer #1 (Comments for the Author):

The authors put together literature and their own data on multi-omics database for *Yersinia* species. I can only applaud this effort, as the transcriptomic data accumulated in this area for the past two decades is quite a mess. Nevertheless, the authors created a workable and useful interactive database where the genomic and phylogenetic data are integrated with transcriptomics and proteomics. It has the user friendly interface and relatively easy to follow. I am confident that this will be a good tool for the *Yersinia* community and others from different areas of bacterial pathogenesis. The planned inclusion in the database genome-wide yeast two-hybrid interactome data, small RNAs, transcriptional start sites, riboswitches, as well as defining the mutants

phenotype (p.15, lines 336-341) is an appropriate direction for further expansion.

Nevertheless, a critical feature is missing in this database. The omics data ideally should be integrated with the biochemical pathways, for example, KEGG Pathways database. Then, genes expressed at certain conditions could be directly linked to the pathways, that can make the dataset truly interactive. This will be particularly important for the proposed defining mutant phenotype. The lack of plans to expand the database to the biochemical pathways significantly reduced the reviewer's enthusiasm for the future of this project.

The minor comments are:

1. p. 6, lines 116-120. To address the phylogeny, the authors used their 500 genes-based cgMLST scheme. I suggest additionally for *Y. pestis* to designate for each strain in the database the phylogenetic branch based on SNP analysis, as this is widely used these days for the phylogenetic relatedness of this pathogen. This should appear on the phylogenetic tree as displayed on the left panel on Figure 3.
2. p.14, line 315. Provide number for the reference Israely et.al.
3. p.16, line 355. "RNA-Seq" or "RNA-Seq", duplication?
4. p.16, lines 367-368. Explain why the tree was rooted on *Y. entomophaga* MH96. This should be briefly defined without a necessity for the reader to go to ref 73.

Reviewer #2 (Comments for the Author):

In this manuscript, Le Bury et. al, have compiled most of the publicly available processed or unprocessed Genomics, Transcriptomics, and Proteomics data of *Yersinia* species in an interactive, user-friendly database called Yersiniomics. The database is constructed on BacNet which was previously used for *Listeria*. The authors have spent a tremendous amount of effort, also time I assume, to build the database. I believe Yersiniomics provides a great source and tool for many researchers in the field of infection biology, especially for those working with different species and even strains of *Yersinia* as a model organism to study bacterial infections. I appreciated the design of the database which allows cross-comparisons of species/strains and different -omics datasets. For example; I highly appreciated that the authors linked the homologues genes in different species/strains which then allows to trace the gene sequence, transcript level, protein level and differential expression in transcript and protein level in different strains and species. Moreover, the interactive feature if the database allows user defined setting for certain tools embedded to the database. Additionally, the addition of new locus tags and old locus tags for each gene helps users to combine the information from this database other types of database which uses either old or the new locus tags. I appreciate that the authors were aware of this confusion and recorded all this information in one place. Finally, the authors indicated the embedding novel datatypes such as yeast two-hybrid interactome data and small RNA data to Yersiniomics, which will give more depth to the database.

Even though I am very much impressed by the idea of constructing Yersiniomics and well-thought details in the design of it, I have some points. These concerns are about the content and analysis of the transcriptomics data and the usage of the database, which I listed below.

The content of the transcriptomics data

The authors claimed that they have used all *Yersinia* omics data published today. They retrieved transcriptomics data for 251 biological conditions, which 151 were originally generated with microarray and retrieved from GEO and 100 were originally generated with RNA-seq and retrieved from ENA. I wonder if the authors are aware of SRA in GEO which, today, contains 644 biosamples (biological conditions) associated to *Yersinia* and generated with RNAseq. Why did not author retrieve this data? They should include this data to Yersiniomics as well. If not, they should have strong evidence about why not doing so.

The analysis of transcriptomics data

- The authors have used RPKM values instead of TPM values as normalized expression level. I would like to know why they preferred RPKM. This could be discussed in the discussion section.
- They have generated Co-expression network using RPKM values with Pearson correlation coefficient via the BacNet platform. Why did they prefer this method while there are well-established Co-expression network construction methods such as WGCNA and ICA? Did authors compare those methods?

The usage of the database

In the Genomics browser,

- The number of replicons is shown as number of chromosomes. It should either have separate columns for chromosome and plasmid or as 'Number of chromosomes/plasmid' which the numbers should be shown as for example; 1/3 (1 chromosome and 3

plasmids.

- For many strains the number of genes, proteins, name of the species and strain, and CladeID is missing. Why are they missing while number of CDSs, rRNA and tRNA consistently exist in all of them?
- When browsing the genome of a particular species, 'Download gene selection as a table' generated an empty txt file named after 'Listeria Genomic Table' even though multiple genes were selected. This should be corrected.
- I could run Synteny function only for once and for a *Yersinia pestis* strain. If possible, it should work for *Y. pseudotuberculosis* and *Y. enterocolitica* also. Does the webpage work equally fine in Windows and MacOS?

In the Transcriptomics browser

- 'Strain array' column is used even for RNA-seq data. The rows with RNA-seq data should have empty cell for this column or indicate 'No applicable'
- In the heatmap transcriptomics part and also at any place in the main text, it is not mentioned what statistical analyses was performed to show the significance of the differential expression. Did the authors employ a p-value or adjusted p-value cut-off? If yes, they should mention in the main text and if not, they should discuss why not.
- When visualizing transcriptomics datasets in Genome viewer and using AddTranscriptomics data, the webpage gives an error and does not allow addition.
- When visualizing transcriptomics datasets in Genome viewer, the webpage does not allow switch from Absolute expression to Relative expression data

Yersiniomics wiki

- Access Yersiniomics wiki directs users to Listeriomics. This should be corrected.

Line 82-86: This sentence should be re-written as it sounds that only Illumina produces short reads and only PacBio produces long reach. There are other technologies producing short and long reads.

Line 355-356. Did the authors specifically download only 'Illumina reads'? If not, they should use 'sequencing reads' instead.

Line 353. 'formatted' to formatted.

Staff Comments:

Preparing Revision Guidelines

Please return the manuscript within 60 days; if you cannot complete the modification within this time period, please contact me. If you do not wish to modify the manuscript and prefer to submit it to another journal, please notify me of your decision immediately so that the manuscript may be formally withdrawn from consideration by Microbiology Spectrum.

In this manuscript, Le Bury et. al, have compiled most of the publicly available processed or unprocessed Genomics, Transcriptomics, and Proteomics data of *Yersinia* species in an interactive, user-friendly database called Yersiniomics. The database is constructed on BacNet which was previously used for *Listeria*. The authors have spent a tremendous amount of effort, also time I assume, to build the database. I believe Yersiniomics provides a great source and tool for many researchers in the field of infection biology, especially for those working with different species and even strains of *Yersinia* as a model organism to study bacterial infections. I appreciated the design of the database which allows cross-comparisons of species/strains and different -omics datasets. For example; I highly appreciated that the authors linked the homologues genes in different species/strains which then allows to trace the gene sequence, transcript level, protein level and differential expression in transcript and protein level in different strains and species. Moreover, the interactive feature if the database allows user defined setting for certain tools embedded to the database. Additionally, the addition of new locus tags and old locus tags for each gene helps users to combine the information from this database other types of database which uses either old or the new locus tags. I appreciate that the authors were aware of this confusion and recorded all this information in one place. Finally, the authors indicated the embedding novel datatypes such as yeast two-hybrid interactome data and small RNA data to Yersiniomics, which will give more depth to the database.

Even though I am very much impressed by the idea of constructing Yersiniomics and well-thought details in the design of it, I have some points. These concerns are about the content and analysis of the transcriptomics data and the usage of the database, which I listed below.

The content of the transcriptomics data

The authors claimed that they have used all *Yersinia* omics data published today. They retrieved transcriptomics data for 251 biological conditions, which 151 were originally generated with microarray and retrieved from GEO and 100 were originally generated with RNA-seq and retrieved from ENA. I wonder if the authors are aware of SRA in GEO which, today, contains 644 biosamples (biological conditions) associated to *Yersinia* and generated with RNAseq. Why did not author retrieve this data? They should include this data to Yersiniomics as well. If not, they should have strong evidence about why not doing so.

The analysis of transcriptomics data

- The authors have used RPKM values instead of TPM values as normalized expression level. I would like to know why they preferred RPKM. This could be discussed in the discussion section.
- They have generated Co-expression network using RPKM values with Pearson correlation coefficient via the BacNet platform. Why did they prefer this method while there are well-established Co-expression network construction methods such as WGCNA and ICA? Did authors compare those methods?

The usage of the database

In the Genomics browser,

- The number of replicons is shown as number of chromosomes. It should either have separate columns for chromosome and plasmid or as 'Number of

chromosomes/plasmid' which the numbers should be shown as for example; 1/3 (1 chromosome and 3 plasmids).

- For many strains the number of genes, proteins, name of the species and strain, and CladeID is missing. Why are they missing while number of CDSs, rRNA and tRNA consistently exist in all of them?
- When browsing the genome of a particular species, 'Download gene selection as a table' generated an empty txt file named after 'Listeria Genomic Table' even though multiple genes were selected. This should be corrected.
- I could run Synteny function only for once and for a *Yersinia pestis* strain. If possible, it should work for *Y. pseudotuberculosis* and *Y. enterocolitica* also. Does the webpage work equally fine in Windows and MacOS?

In the Transcriptomics browser

- 'Strain array' column is used even for RNA-seq data. The rows with RNA-seq data should have empty cell for this column or indicate 'No applicable'
- In the heatmap transcriptomics part and also at any place in the main text, it is not mentioned what statistical analyses was performed to show the significance of the differential expression. Did the authors employ a p-value or adjusted p-value cut-off? If yes, they should mention in the main text and if not, they should discuss why not.
- When visualizing transcriptomics datasets in Genome viewer and using AddTranscriptomics data, the webpage gives an error and does not allow addition.
- When visualizing transcriptomics datasets in Genome viewer, the webpage does not allow switch from Absolute expression to Relative expression data

Yersiniomics wiki

- Access Yersiniomics wiki directs users to Listeriomics. This should be corrected.

Line 82-86: This sentence should be re-written as it sounds that only Illumina produces short reads and only PacBio produces long reach. There are other technologies producing short and long reads.

Line 355-356. Did the authors specifically downloaded only 'Illumina reads'? If not, they should use 'sequencing reads' instead.

Line 353. 'formated' to formatted.

Reviewer #1

The authors put together literature and their own data on multi-omics database for *Yersinia* species. I can only applaud this effort, as the transcriptomic data accumulated in this area for the past two decades is quite a mess. Nevertheless, the authors created a workable and useful interactive database where the genomic and phylogenetic data are integrated with transcriptomics and proteomics. It has the user friendly interface and relatively easy to follow. I am confident that this will be a good tool for the *Yersinia* community and others from different areas of bacterial pathogenesis. The planned inclusion in the database genome-wide yeast two-hybrid interactome data, small RNAs, transcriptional start sites, riboswitches, as well as defining the mutants phenotype (p.15, lines 336-341) is an appropriate direction for further expansion.

We thank the reviewer for highlighting the interest of our database.

Nevertheless, a critical feature is missing in this database. The omics data ideally should be integrated with the biochemical pathways, for example, KEGG Pathways database. Then, genes expressed at certain conditions could be directly linked to the pathways, that can make the dataset truly interactive. This will be particularly important for the proposed defining mutant phenotype. The lack of plans to expand the database to the biochemical pathways significantly reduced the reviewer's enthusiasm for the future of this project.

We thank the reviewer for this suggestion. For most reference strains, we have now added tabs in the gene viewer with dynamic links to KEGG, UniProt, InterPro, IntAct and STRING (p.13, lines 278-288 of the final manuscript in PDF format). These tabs allow to directly browse this database inside Yersiniomics and are automatically updated to the entry of the selected gene. These new functionalities will help decipher gene functionalities, crossing known structural or functional data and pathways to experimental results performed on *Yersinia*.

1. p. 6, lines 116-120. To address the phylogeny, the authors used their 500 genes-based cgMLST scheme. I suggest additionally for *Y. pestis* to designate for each strain in the database the phylogenetic branch based on SNP analysis, as this is widely used these days for the phylogenetic relatedness of this pathogen. This should appear on the phylogenetic tree as displayed on the left panel on Figure 3.

Following the reviewer's suggestion, we added "lineage" and "sublineage" columns in the genomics browser (p.7 line 137), in which lineages were determined by cgMLST and *Yersinia pestis* sublineages were determined by SNP analysis (p.7 lines 137-139).

2. p.14, line 315. Provide number for the reference Israely et.al.

We added this reference (p.18 line 399).

3. p.16, line 355. "RNA-Seq" or "RNA-Seq", duplication?

We corrected this in " 'RNA-Seq' or 'RNASeq' " (p.20 line 443)

4. p.16, lines 367-368. Explain why the tree was rooted on *Y. entomophaga* MH96. This should be briefly defined without a necessity for the reader to go to ref 73.

We rooted the tree on *Y. entomophaga* as it is the most ancestral branch of the *Yersinia* genus (p.21 line 457-458).

Reviewer #2

In this manuscript, Le Bury et. al, have compiled most of the publicly available processed or unprocessed Genomics, Transcriptomics, and Proteomics data of *Yersinia* species in an interactive, user-friendly database called Yersiniomics. The database is constructed on BacNet which was previously used for *Listeria*. The authors have spent a tremendous amount of effort, also time I assume, to build the database. I believe Yersiniomics provides a great source and tool for many researchers in the field of infection biology, especially for those working with different species and even strains of *Yersinia* as a model organism to study bacterial infections. I appreciated the design of the database which allows cross-comparisons of species/strains and different -omics datasets. For example; I highly appreciated that the authors linked the homologues genes in different species/strains which then allows to trace the gene sequence, transcript level, protein level and differential expression in transcript and protein level in different strains and species. Moreover, the interactive feature if the database allows user defined setting for certain tools embedded to the database. Additionally, the addition of new locus tags and old locus tags for each gene helps users to combine the information from this database other types of database which uses either old or the new locus tags. I appreciate that the authors were aware of this confusion and recorded all this information in one place. Finally, the authors indicated the embedding novel datatypes such as yeast two-hybrid interactome data and small RNA data to Yersiniomics, which will give more depth to the database.

Even though I am very much impressed by the idea of constructing Yersiniomics and well-thought details in the design of it, I have some points. These concerns are about the content and analysis of the transcriptomics data and the usage of the database, which I listed below.

We thank the reviewer for stressing our effort and appreciating the functionalities of our database.

The content of the transcriptomics data

The authors claimed that they have used all *Yersinia* omics data published today. They retrieved transcriptomics data for 251 biological conditions, which 151 were originally generated with microarray and retrieved from GEO and 100 were originally generated with RNA-seq and retrieved from ENA. I wonder if the authors are aware of SRA in GEO which, today, contains 644 biosamples (biological conditions) associated to *Yersinia* and generated with RNAseq. Why did not author retrieve this data? They should include this data to Yersiniomics as well. If not, they should have strong evidence about why not doing so.

We thank the reviewer for this comment and we have now made substantial complements to our analysis. When searching for RNA experiment on the SRA database (<https://www.ncbi.nlm.nih.gov/sra>) with the keyword "*Yersinia*", we can indeed retrieve 801 experiments (09.01.2023). Most of the recorded experiments include only one run consisting of individual replicates in one biological condition, but some authors submitted data including several runs, consisting of several replicates, in one experiment. When searching for individual RNA-Seq runs via SRA Run Selector (<https://www.ncbi.nlm.nih.gov/Traces/study>) using the keyword "*Yersinia*", we could retrieve 907 runs. We have now analyzed 426 of these runs, which represent 166 biological

conditions when aggregating replicates (previously, 100 biological conditions were analyzed in our original manuscript). The majority of RNA-Seq runs available on SRA were processed with the exception of 325 runs sequencing infected- or uninfected host RNAs, 84 runs aiming at identifying transcriptional start sites, 39 runs sequencing phages and 33 last runs lacking metadata or failing quality control.

The analysis of transcriptomics data

- The authors have used RPKM values instead of TPM values as normalized expression level. I would like to know why they preferred RPKM. This could be discussed in the discussion section.

TPM is indeed widely used and facilitates comparisons, as the total count is normalized to 1 million. We therefore changed our pipeline and calculated TPM instead of RPKM (p.12 line 262-265 of the final manuscript in PDF format).

- They have generated Co-expression network using RPKM values with Pearson correlation coefficient via the BacNet platform. Why did they prefer this method while there are well-established Co-expression network construction methods such as WGCNA and ICA? Did authors compare those methods?

The co-expression networks were initially calculated using all RNA-Seq data for each genome. Given: a) the absence of consistency between different experiments based on the same genome, b) the high number of networks we could compute and c) the fact that networks were heavy to load on the website in its initial form, we decided to remove this functionality from Yersiniomics. As an alternative, users can now download the TPM normalized data of all biological conditions of a specified genome in an excel file (p.14 lines 290-300), and compute co-expression networks using conditions of their choice.

The usage of the database

In the Genomics browser,

- The number of replicons is shown as number of chromosomes. It should either have separate columns for chromosome and plasmid or as 'Number of chromosomes/plasmid' which the numbers should be shown as for example; 1/3 (1 chromosome and 3 plasmids).

We created two columns, one with the chromosome number and one for the number of plasmids (Figure 3).

- For many strains the number of genes, proteins, name of the species and strain, and CladeID is missing. Why are they missing while number of CDSs, rRNA and tRNA consistently exist in all of them?

Most of these columns (genes, proteins, cladeID) were relics of previous unmaintained code. To homogenize the results, we retained only the "CDS" and "rRNA and tRNA" columns computed from the annotation .GFF file downloaded from GenBank and used to construct the database. The "species" and "strain" columns were updated.

- When browsing the genome of a particular species, 'Download gene selection as a table' generated and empty txt file named after 'Listeria Genomic Table' even though multiple genes

were selected. This should be corrected.

This bug was corrected. Gene selection can now be downloaded as a table name 'Yersinia Genomic Table'.

- I could run Synteny function only for once and for a Yersinia pestis strain. If possible, it should work for Y. pseudotuberculosis and Y. enterocolitica also. Does the webpage work equally fine in Windows and MacOS?

This bug was due to a synchronization problem between the Yersiniomics software written in Java and the SynTView software written in JavaScript. This bug is now corrected when displaying the synteny with the "show synteny" button (p.11 lines 227-229). Depending on the network connection, loading can require a bit of time before the synteny appears. The webpage works in both Windows and MacOS.

In the Transcriptomics browser

- 'Strain array' column is used even for RNA-seq data. The rows with RNA-seq data should have empty cell for this column or indicate 'No applicable'

"Strain array" was replaced by "Reference strain" as it consists in an array for microarray data, and in a mapped genome for RNA-Seq experiments.

- In the heatmap transcriptomics part and also at any place in the main text, it is not mentioned what statistical analyses was performed to show the significance of the differential expression. Did the authors employ a p-value or adjusted p-value cut-off? If yes, they should mention in the main text and if not, they should discuss why not.

We could not consistently retrieve p-values for microarray data. However, we could calculate the p-values and adjusted p-values for all RNA-Seq experiments with replicates, using DESeq2 default tests (p.22 lines 489-498). We also calculated p-values and adjusted p-values for the proteomes when we retrieved the raw intensities and calculated the fold changes ourselves (p. 22-23 lines 499-513). When we could not retrieve any p-value from publications, or in absence of replicates, we set the p-value to 0.

All these p-values and adjusted p-values are now displayed next to fold changes in the gene viewer (p.12 line 256-258 and p.13 lines 271-273, figure 9). We added a p-value cut-off allowing to filter the up- and down-regulated datasets (p.12 line 258 and p.13 lines 271-273, figure 9). Moreover, the DESeq2 reports of RNA-Seq experiments allow to filter and sort tables of fold changes and to download the results (p.10 lines 197-212)

- When visualizing transcriptomics datasets in Genome viewer and using AddTranscriptomics data, the webpages gives an error and does not allow addition.

This bug was corrected. Other transcriptomics or proteomics data can now be added to the genome viewer from Yersiniomics database.

- When visualizing transcriptomics datasets in Genome viewer, the webpage does not allow switch from Absolute expression to Relative expression data

This bug was corrected. When no absolute data exists (for microarray for example), data remain in relative mode, while other data (such as mass spectrometry or RNA-Seq data) switch to absolute mode.

- Access Yersiniomics wiki directs users to Listeriomics. This should be corrected.

We removed this button and added a link to the article (to be updated after the publication) on the homepage of Yersiniomics, to guide users requiring more information on the website use.

Line 82-86: This sentence should be re-written as it sounds that only Illumina produces short reads and only PacBio produces long reach. There are other technologies producing short and long reads.

This sentence was re-written (p.5 lines 87-92).

Line 355-356. Did the authors specifically downloaded only 'Illumina reads'? If not, they should use 'sequencing reads' instead.

At the date of the first submission, the 100 RNA-Seq biological conditions consisted only in Illumina sequencing project. However, the 166 RNA-Seq biological conditions we added to date used Illumina sequencing as well as BGISEQ-500 or AB 5500 platform. We changed the sentence as stated by the reviewer (p.20 lines 445-446)

Line 353. 'formated' to formatted.

This has been corrected (p.20 line 441).

January 26, 2023

Dr. Javier Pizarro-Cerdá
Institut Pasteur
Unité de Recherche Yersinia
Paris
France

Re: Spectrum03826-22R1 (Yersiniomics, a multi-omics interactive database for *Yersinia* species)

Dear Dr. Javier Pizarro-Cerdá:

Your manuscript has been accepted, and I am forwarding it to the ASM Journals Department for publication. You will be notified when your proofs are ready to be viewed.

Sincerely,
Tino Polen
Editor, Microbiology Spectrum
